# Heterodimerization of Munc13 $C_2A$ domain with RIM regulates synaptic vesicle docking and priming

Marcial Camacho[1,2,3], Jayeeta Basu[3,†], Thorsten Trimbuch[1,2], Shuwen Chang[1,2], Cristina Pulido-Lozano[1,2], Shwu-Shin Chang[4], Irina Duluvova[4], Masin Abo-Rady[1,2], Josep Rizo[4] & Christian Rosenmund[1,2,3]

The presynaptic active zone protein Munc13 is essential for neurotransmitter release, playing key roles in vesicle docking and priming. Mechanistically, it is thought that the $C_2A$ domain of Munc13 inhibits the priming function by homodimerization, and that RIM disrupts the autoinhibitory homodimerization forming monomeric priming-competent Munc13. However, it is unclear whether the $C_2A$ domain mediates other Munc13 functions in addition to this inactivation–activation switch. Here, we utilize mutations that modulate the homodimerization and heterodimerization states to define additional roles of the Munc13 $C_2A$ domain. Using electron microscopy and electrophysiology in hippocampal cultures, we show that the $C_2A$ domain is critical for additional steps of vesicular release, including vesicle docking. Optimal vesicle docking and priming is only possible when Munc13 heterodimerizes with RIM via its $C_2A$ domain. Beyond being a switching module, our data suggest that the Munc13-RIM heterodimer is an active component of the vesicle docking, priming and release complex.

[1] Institute of Neurophysiology, Charité—Universitätsmedizin Berlin, 10117 Berlin, Germany. [2] NeuroCure Cluster of Excellence, Charité—Universitätsmedizin Berlin, 10117 Berlin, Germany. [3] Department of Neuroscience, Baylor College of Medicine, Houston, Texas 77030, USA. [4] Departments of Biophysics, Biochemistry and Pharmacology, University of Texas Southwestern Medical Center, Dallas, Texas 75390, USA. † Present address: Department of Neuroscience and Physiology, NYU Neuroscience Institute, New York University School of Medicine, New York, New York 10016, USA. Correspondence and requests for materials should be addressed to C.R. (email: christian.rosenmund@charite.de).

nformation processing in the central nervous system relies on efficient synaptic transmission. Neurotransmitter (NT) release occurs at a specialized presynaptic release site, known as the active zone (AZ), and depends on the availability of docked and primed fusion-competent vesicles. The processes of docking, priming and fusion rely on the function of specific proteins at the AZ. Loss-of-function studies in *Caenorhabditis elegans*, *Drosophila melanogaster* and mammalian synapses have identified several AZ protein families that play a key role in vesicle docking and priming, including Munc13 (refs 1–6), CAPS[4,7], α-Liprin[8], ELKS[9] and RIM[2,10–13].

Among these proteins, Munc13s and their homologues in *C. elegans* and *D. melanogaster* are of particular interest because they are absolutely crucial for both of these steps in the synaptic vesicle (SV) cycle. In the mammalian central nervous system, the Munc13 protein family comprises of four members: the most prominently expressed Munc13-1, the two Munc13-2 isoforms bMunc13-2 and ubMunc13-2, and Munc13-3 (ref. 14). Neurons from *Munc13-1/2* double knockout (DKO) lack of a measurable ready releasable pool (RRP) and show no spontaneous or $Ca^{2+}$-dependent evoked release[1,6], similar to the observations made in the corresponding *unc-13* mutant in *C. elegans*[15] and in *D. melanogaster dunc-13* mutant[16]. This absence of neurotransmission was initially attributed to events downstream of SV docking, based on ultrastructural data from conventional chemical fixation[1]. However, reexamination of Unc13/Munc13 deficiency in *C. elegans*[2,17,18] and mice[3–5] using high-pressure freeze fixation revealed a loss of docked SVs, indicating that Munc13 plays a role in the vesicle docking mechanism.

All Munc13s expressed at central synapses are large multidomain proteins and share a well-conserved C-terminal region, including a diacylglycerol (DAG) binding $C_1$ domain, a $Ca^{2+}$/phospholipid binding $C_2B$ domain, a MUN domain and a $C_2C$ domain at the very C-terminal end. Biophysical studies show that the Munc13 C terminus is crucial for membrane fusion in reconstitution experiments that include the SNAREs, Munc18-1, NSF (*N*-ethylmaleimide-sensitive factor) and α-SNAP (α-soluble NSF attachment protein) because Munc13 and Munc18-1 orchestrate SNARE complex assembly in an NSF/α-SNAP-resistant manner[19]. Structure–function analyses revealed that the C-terminal MUN domain is essential for vesicle priming activity[20–22] and mediates the activity of Munc13 by opening syntaxin-1 (refs 23–25). Moreover, Munc13-1 fragment including the $C_1C_2B$ region preceding the MUN domain and the $C_2C$ domain is critical for neurotransmitter release as they may help bridge the SV and plasma membrane, that in turn facilitates opening of syntaxin-1 (ref. 26). In addition, the $C_1$ (refs 27–29) and $C_2B$ domain[30] are important regulators of vesicle release probability and short-term plasticity, modulating vesicle fusion efficiency through interaction with the plasma membrane in a DAG- or $Ca^{2+}$/phospholipid-dependent manner.

The Munc13 N terminus, on the other hand, varies considerably between the Munc13 isoforms[14]. In *C. elegans*, the long and short Unc13 isoforms differ in their regulation of both vesicle location relative to $Ca^{2+}$ channels and neurotransmitter release kinetics[31]. In mammals, the N termini of the isoforms Munc13-1 and ubMunc13-2 contain a calmodulin (CaM)-binding sequence that regulates short-term plasticity[32,33], and a $C_2A$ domain that is not present in bMunc13-2 and Munc13-3. Biochemical studies have demonstrated that the $C_2A$ domain can form tight homodimeric complexes[34] or alternatively, a heterodimeric complex with the zinc-finger domain (ZF) of the RIM protein that can bind to the SV protein Rab3 (ref. 35). Based on functional studies of Munc13 $C_2A$ domain interactions, two distinct roles have been proposed for this domain: the recruitment of Munc13 to the presynaptic active zone[36] and the

activation of Munc13 as prelude to vesicle priming activity[37]. In the latter study, Munc13 homodimers were shown to be inactive and priming incompetent, while competitive binding of RIM to the $C_2A$ disrupted the Munc13 homodimer and activated vesicle priming. Remarkably, either expression of an N-terminally truncated Munc13 or a Munc13 with a $C_2A$ domain point mutation that inhibits Munc13 homodimerization in *RIM1/2* conditional DKO bypassed the requirement of RIM in vesicle priming, suggesting that RIM only interacts with Munc13 to disrupt the homodimer, and indicating that the active form of Munc13 is the monomer. These findings further implied that the Munc13/RIM complex does not play a role in priming itself and that the Munc13 $C_2A$ domain merely acts as an inactivation–activation switch. This model is surprising given that classical electron microscopy (EM) studies of aldehyde-fixed synapses lacking all RIM1 and RIM2 isoforms[12,13] and EM tomography from high-pressure freeze synapses of *unc-10* mutant in *C. elegans*[2,17,18] show a reduction in docked vesicles. Moreover, the Munc13/RIM/Rab3 complex may play a direct role in regulating release probability[38]. On the other hand, it is unclear how the conversion of Munc13 from homodimers to monomers or Munc13/RIM heterodimers affects other steps in the SV cycle: does it only affect priming or also upstream processes, such as vesicle docking, and downstream processes, such as release? Does homodimerization have a functional role beyond maintaining Munc13 in an inactive state? To clarify the role of the $C_2A$ domain and the individual functions of Munc13 monomers, Munc13/Munc13 homodimers and Munc13/RIM heterodimers, we performed a structure–function analysis of the role of the $C_2A$ domain of Munc13-1 in docking, priming and vesicular release in mammalian central synapses.

## Results

**Munc13-1 $C_2A$ domain supports vesicle docking.** The complete loss of docked synaptic vesicles observed in electron tomography and transmission electron microscopy (TEM) studies using cryofixation techniques on *Munc13-1/2* DKO neurons[3–5] clearly demonstrate that Munc13s are involved in SV docking. The interaction between the Munc13-1 $C_2A$ domain and the RIM ZF domain[39] and the proposed role of RIM in vesicle docking[2,11–13] suggest that the function of Munc13-1 in vesicle docking could be mediated by the interaction of the $C_2A$ domain with RIM. To test this hypothesis, we performed gain-of-function rescue experiments in *Munc13-1/2* DKO cultures from murine hippocampal neurons. Using lentiviral gene transfer, we expressed three different mutants of Munc13-1. The first was Munc13-1 with a complete truncation of the N terminus (Munc13-1 del 1–520), corresponding to the mutant published previously[37,39]. The other two mutants were designed to test the specific $C_2A$ domain function within the N terminus: one that specifically eliminated the $C_2A$ domain and the subsequent α-helix (Munc13 del 1–150), and another that incorporated the $C_2A$ domain but eliminated the intermediate region spanning from the end of the $C_2A$ domain to the region preceding the $C_1$ domain (Munc13-1 del 151–520) (Fig. 1a). *Munc13-1/2* DKO neurons rescued with a full-length Munc13-1 protein (Munc13-1 wild type (WT)) served as our WT-positive control and untransfected *Munc13-1/2* DKO neurons served as negative controls. All Munc13-1 deletion mutants were found at presynaptic glutamatergic compartments, as indicated by the green fluorescent protein (GFP) signal detected at the vesicular glutamate transporter 1 (VGLUT1)-positive synapses (Supplementary Fig. 1a). Fluorescence intensity quantification of the GFP signal at the VGLUT1 compartments (Supplementary Fig. 1b) showed that all three Munc13-1 deletion mutants were

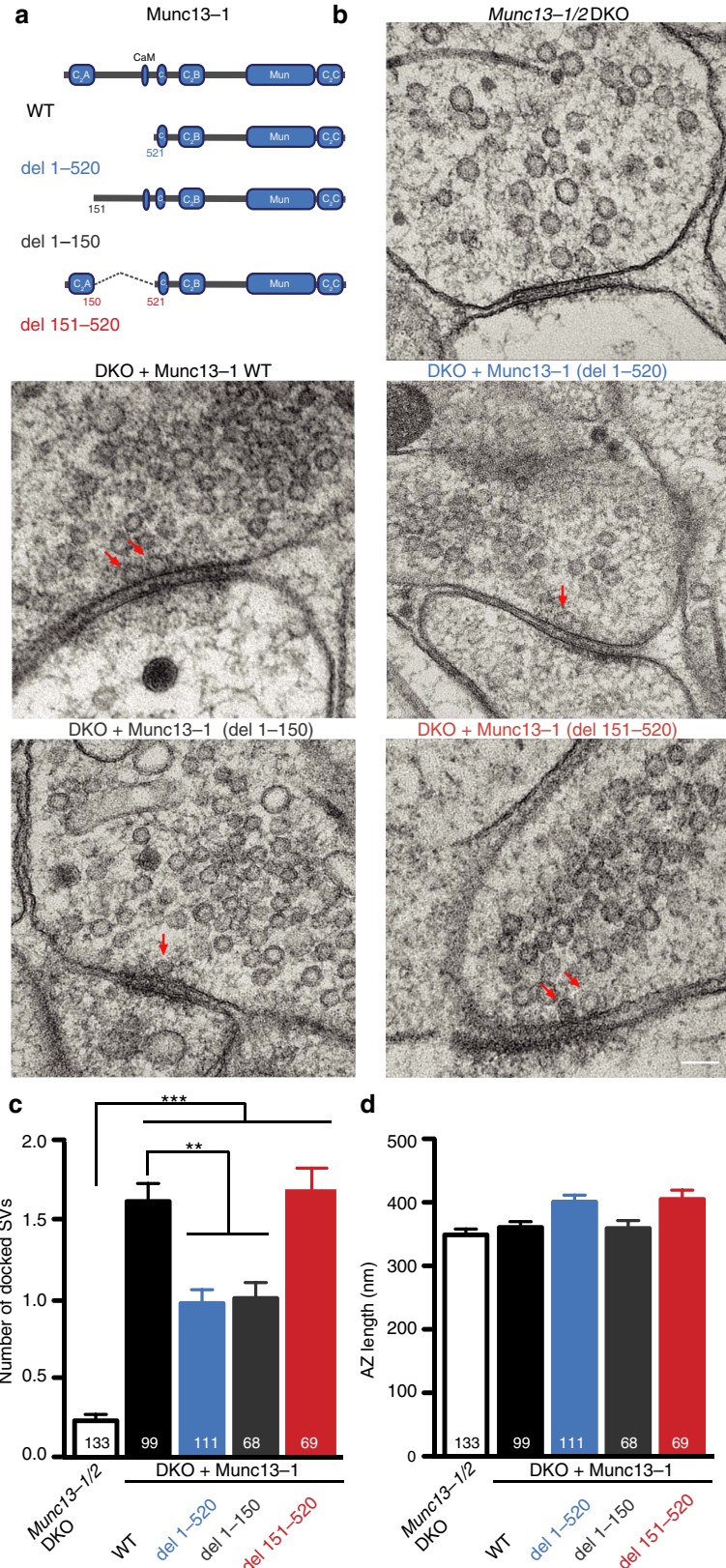

**Figure 1 | The N-terminal C₂A domain of Munc13-1 is necessary for proper docking of synaptic vesicles.** (**a**) Domain structure of full-length Munc13-1 and mutant proteins rescued in *Munc13-1/2* DKO hippocampal neurons. (**b**) Representative electron micrographs of synapses showing docked synaptic vesicles (indicated by red arrows) from *Munc13-1/2* DKO hippocampal cultures and DKO cultures rescued with the respective Munc13-1 WT and mutants. Scale bar, 100 nm. (**c**) Plot of number of docked SVs. (**d**) Plot of AZ length in nm. Numbers in bar graphs represent the *n* values for each group. Significances and *P* values were determined by one-way analysis of variance (ANOVA) with Kruskal–Wallis test followed by Dunn's post test. Values indicate mean ± s.e.m.; **$P < 0.01$, ***$P < 0.001$.

expressed at similar or higher levels than WT. Furthermore, the comparison of synaptic responses between $Munc13\text{-}1^{+/+}/Munc13\text{-}2^{-/-}$, $Munc13\text{-}1^{+/-}/Munc13\text{-}2^{-/-}$ and $Munc13\text{-}1^{+/+}/Munc13\text{-}2^{-/-}$ overexpressing Munc13-1 WT neurons did not reveal significant changes in $Ca^{2+}$-evoked release or

priming activity, indicating that these synaptic properties do not change when expression levels vary by ~50% of WT levels (Supplementary Fig. 2d). We also determined the degree of colocalization of Munc13 and VGLUT1 signals by comparing Pearson's correlation index in the three truncated mutants.

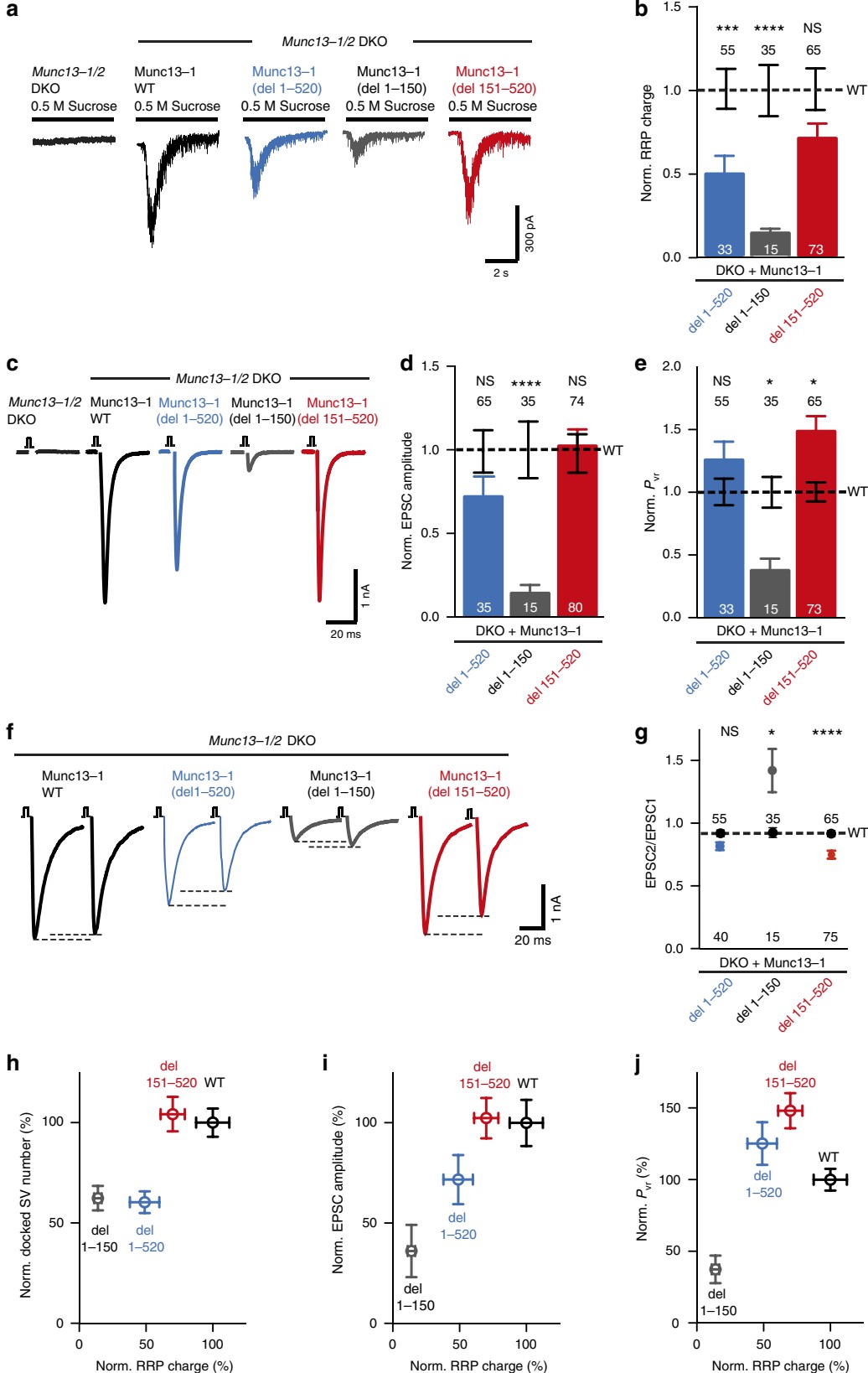

Supplementary Fig. 1c shows that the three truncated mutants differ only slightly from the WT and between each other, although the mutants that lacked $C_2A$ domain tended to show more diffuse signals in axons and neurites (Supplementary Fig. 1a). The efficiency of the *Munc13-1/2* DKO rescue was robust, as synaptic responses reached 65% of the $Ca^{2+}$-evoked release and 75% of the priming activity compared with the *Munc13-1$^{+/+}$/Munc13-2$^{-/-}$* neurons (Supplementary Fig. 2c).

To address the role of Munc13 $C_2A$ domain in vesicle docking, we performed TEM on high-pressure frozen samples from the *Munc13-1/2* DKO and *Munc13-1/2* DKO rescued with Munc13-1 WT or with the Munc13-1 $C_2A$ domain-deleted mutants. EM analysis was performed on 40 nm sections and the number of synaptic vesicles in physical contact with the plasma membrane at active zones was counted. In line with previous studies[3–5], *Munc13-1/2* DKO active zones were essentially devoid of docked vesicles ($0.2 \pm 0.04$ docked vesicle/AZ), while the active zones of the Munc13-1 WT-rescued synapses showed approximately 2 docked synaptic vesicles ($1.62 \pm 0.11$ docked vesicle/AZ). However, synapses from DKO rescued with the two Munc13-1 mutants that lacked the $C_2A$ domain, that is, the $C_2A$ domain (del 1–150) and the N-terminus (del 1–520) truncations, showed a 42% reduction in the number of docked SVs. In contrast, deletion of the intermediate region of the N terminus that does not eliminate the $C_2A$ domain (del 151–520) showed the same docking activity as the Munc13-1 WT (Fig. 1b,c). We did not detect differences in the number of SVs within 100 nm of the AZ, in the diameter of the SVs or in AZ length between any of the groups (Fig. 1d and Supplementary Table 1), demonstrating that synapse formation was unaffected. We conclude that the $C_2A$ domain of Munc13-1 is required to maintain the full-sized pool of docked vesicles, indicating an important positive modulation for docked synaptic vesicles.

**Munc13 $C_2A$ domain is critical for vesicle priming and release.** Impairment of SV docking is thought to negatively affect downstream functions such as vesicle priming, a process of making synaptic vesicle fusion competent and $Ca^{2+}$-triggered release. To correlate docking to priming and fusion, we assayed vesicle priming and release in the Munc13-1 mutants using electrophysiological recordings in autaptic hippocampal cultures[40]. Vesicle priming was estimated by determining the size of the RRP. The RRP was quantified by integrating the transient inward current induced by a 5 s pulsed application of hypertonic sucrose solutions[41].

As shown previously[1], *Munc13-1/2* DKO neurons neither released neurotransmitter upon application of hypertonic sucrose solution (left trace, Fig. 2a), nor exhibited measurable synaptic output after action potential stimulation (left trace, Fig. 2c). None of the deletion mutants fully rescued vesicle priming and had differential effects on the action potential (AP)-evoked excitatory postsynaptic current (EPSC). The Munc13-1 mutant that conserves the $C_2A$ domain but lacks the intermediate

N-terminal region (del 151–520) only moderately altered RRP size without affecting the AP-evoked EPSC. The Munc13-1 mutant that lacked the entire N terminus (del 1–520) caused a 50% reduction in RRP size, similar in extent to the reduction in vesicle docking, but only a mildly altered AP-evoked response. However, the mutant that lacked the $C_2A$ domain and the subsequent α-helix (del 1–150) showed a severe 80% loss of primed vesicles and evoked synaptic output (Fig. 2a–d).

The differential effects in vesicle priming and AP-evoked release in several of the mutants suggests an additional role of the Munc13 N terminus in neurotransmitter release probability. Indeed, deletion of the intermediate region of the N terminus (del 151–520) led to an increase in vesicular release probability ($P_{vr}$), while truncation of the $C_2A$ domain (del 1–150) caused a reduction in $P_{vr}$. Deleting both the $C_2A$ domain and the intermediate region from the N terminus (del 1–520) exhibited a relative increase in $P_{vr}$ but did not reach statistical significance (Fig. 2e). Consistent with the observed alterations in vesicular release probability, we found that truncation of the $C_2A$ domain led to strong facilitation, while the Munc13-1 mutant lacking the complete N terminus or the intermediate region displayed enhanced depression compared with full-length Munc13-1 (Fig. 2f,g).

Overall, our functional analyses of the Munc13-1 deletion mutants lead to two major findings. Firstly, the $C_2A$ domain plays an important role in vesicle priming and release probability. Secondly, the Munc13-1 N terminus must contain domains that have both inhibitory and facilitatory effects on vesicle priming and release probability, since the short truncation of the N terminus (amino acids 1–150), including the $C_2A$ domain and the subsequent α-helix, is not functionally equivalent to the truncation of the entire N terminus (amino acids 1–520) (Fig. 2i,j). This raises the hypothesis that the Munc13-1 N terminus has other functions in addition to the $C_2A$/RIM-dependent activation switch.

**Disruption of the homo- and heterodimeric complex of Munc13.** Biochemical and functional studies have shown that the Munc13-1 $C_2A$ monomers can interact either with another Munc13 $C_2A$ domain to form a homodimer or alternatively with RIM to form a heterodimeric complex[34]. Given the multiple functions of the $C_2A$ domain of Munc13 in vesicle docking, priming and release probability, we asked whether the Munc13 $C_2A$ domain functions differentially depending on its interaction partner. Indeed, while the Munc13 homodimer is thought to inhibit priming, disruption of the Munc13 homodimer via RIM binding to the $C_2A$ domain activates priming[37]. To further examine the consequences of Munc13 $C_2A$ domain interactions on vesicle docking, priming and release probability, we generated a set of point mutants in the Munc13-1 $C_2A$ domain in which homodimerization and/or RIM heterodimerization is disrupted (Fig. 3a). The K32E single point mutant is known to disrupt Munc13-$C_2A$ homodimerization without disrupting Munc13-$C_2A$/ZF-RIM heterodimerization[34].

**Figure 2 | Impairment of RRP, EPSC and Pvr on truncation of the C2A domain in Munc13-1.** (**a**) Representative traces of synaptic responses induced by 500 mM sucrose from *Munc13-1/2* DKO autaptic hippocampal cultures and DKO cultures rescued with the respective Munc13-1 WT and mutants indicated above. (**b**) Plot of RRP charge of Munc13-1 mutants normalized (Norm.) to corresponding Munc13-1 WT data. (**c**) Representative traces of AP-evoked EPSC amplitudes recorded in autaptic hippocampal neurons from *Munc13-1/2* DKO and DKO rescued with Munc13-1 mutants indicated above. (**d**) Plot of AP-evoked EPSC amplitudes of Munc13-1 mutants normalized to corresponding Munc13-1 WT. (**e**) Plot of $P_{vr}$ of Munc13-1 mutants normalized to corresponding Munc13-1 WT. (**f**) Example traces of EPSC amplitudes in response to 2 APs separated by 100 ms (10 Hz) of DKO rescued with Munc13-1 WT and mutants indicated above. (**g**) Graph showing average paired-pulse ratios calculated from the 2 AP-evoked EPSC amplitudes. WT rescue is shown as black dotted line. (**h–j**) Correlation between primed synaptic vesicle and docked synaptic vesicle, AP-evoked EPSC amplitudes and vesicular release probability from DKO neurons rescued with Munc13-1 WT and N-terminal deletion mutants. Numbers in plots are *n* values for each mutant group and numbers above the dashed line are the corresponding WT *n* numbers. Error bars represent s.e.m. Significances and *P* values were determined by one-way analysis of variance (ANOVA) with Kruskal–Wallis test followed by Dunn's post test. Values indicate mean ± s.e.m.; *$P < 0.05$; ***$P < 0.001$; ****$P < 0.0001$.

To specifically disrupt Munc13-$C_2$A/ZF-RIM heterodimerization without affecting the homodimerization state, we designed a double point mutation within the $C_2$A domain (E128K) and the subsequent α-helix (E137K) based on the crystal structure of the Munc13-1 $C_2$A domain bound to the RIM ZF domain[34] (Fig. 3b). Isothermal titration calorimetry (ITC)

showed that a WT Munc13-1 fragment containing the $C_2$A domain and the additional α-helix (residues 3–150) binds robustly to the RIM ZF domain, while the Munc13-1 fragment (3–150) with the double point mutation (E128K, E137K) exhibited no detectable binding (Fig. 3c). Finally, a triple point mutant combining the two previous mutations (K32E, E128K,

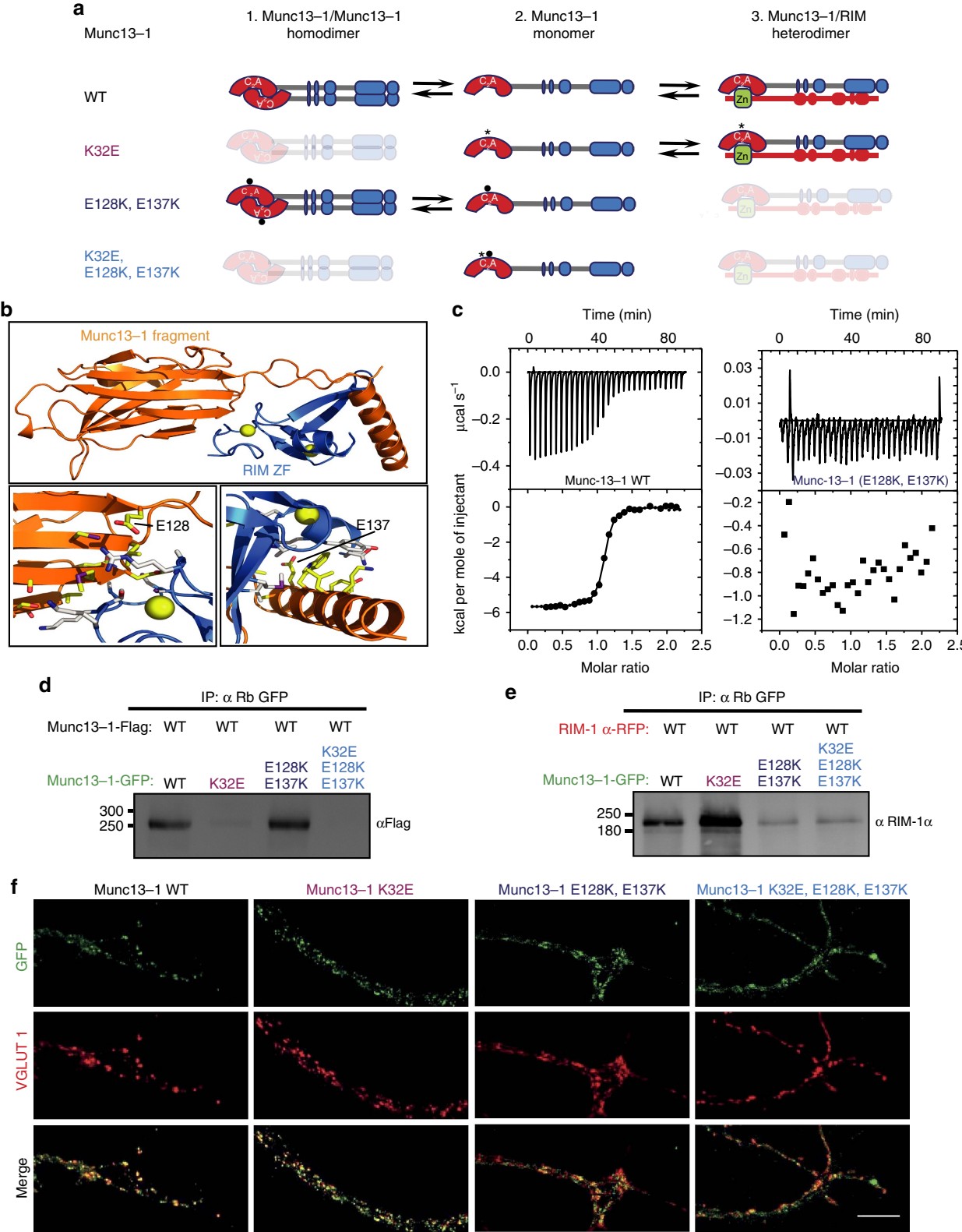

E137K) was created to study the Munc13-1 function as monomer (Fig. 3a).

We confirmed the impairment of binding in these Munc13-1 $C_2A$ mutations using co-immunoprecipitation assays in HEK293 cells. To assess Munc13 homodimerization, we co-transfected a C-terminally flag-tagged Munc13-1 with Munc13-1 WT or each of the three $C_2A$ point mutants, fused C terminally with the GFP. The two homodimerization-disrupting mutants Munc13-1 (K32E) and Munc13-1 (K32E, E128K, E137K) failed to pull down flag-tagged Munc13-1, demonstrating that they do not form $C_2A/C_2A$ homodimers (Fig. 3d and Supplementary Fig. 3a). To probe for Munc13-$C_2A$/ZF-RIM heterodimerization, we co-transfected HEK293 cells with Munc13-1 WT or the three $C_2A$ point mutants GFP-tagged together with full-length RIM-1α fused C terminally to a red fluorescent protein (RFP). As expected, Munc13-1 WT and the Munc13-1 (K32E) that favours the heterodimerization state efficiently co-immunoprecipitated RIM-1α. However, the two Munc13-$C_2A$/ZF-RIM heterodimerization-disrupting mutants ((E128K, E137K) and (K32E, E128K, E137K)) exhibited a major disruption to RIM binding (Fig. 3e and Supplementary Fig. 3b).

To determine the synaptic localization and expression of the Munc13-1 rescue constructs, we analysed and quantified their GFP signal at VGLUT1-positive compartments. The results showed that all expressed mutant proteins exhibited presynaptic localization (Fig. 3f and Supplementary Fig. 4). Quantification of the intensities of the GFP fluorescence at VGLUT1-positive compartments showed that the homodimerization-disrupting mutants Munc13-1 (K32E) and (K32E, E128K, E137K) have similar or higher expression levels than the WT, while the heterodimerization-disrupting mutant Munc13-1 (E128K, E137K) showed an ∼30% reduction in expression (Supplementary Fig. 4b). The degree of colocalization of the mutant proteins with VGLUT1 showed slightly increased colocalization for the homodimerization-disrupting mutants Munc13-1 (K32E), while the heterodimerization-disrupting mutants Munc13-1 (E128K, E137K) and (K32E, E128K, E137K) displayed decreased colocalization. Accordingly, the latter two mutants tended to show more diffuse signals in axons and neurites (Fig. 3f).

Our observation of the localization of the three protein mutants at the presynaptic glutamatergic compartments suggest that Munc13-1 can in principle co-exist as homodimer, monomer and heterodimer with RIM. We then tested how the shift in the equilibrium between these three states could affect their synaptic function. The single point mutation (K32E) helps to study the role of Munc13-1 in the Munc13/RIM heterodimeric state, while the double point mutation (E128K, E137K) probes the function of the Munc13-1 as a homodimer. The triple point mutation (K32E, E128K, E137K) represents the monomeric state (Fig. 3a).

**Optimal SV docking requires Munc13 RIM heterodimer formation.** To assess the impact of the different states of Munc13 on vesicle docking, we analysed EMs of *Munc13-1/2* DKO synapses expressing the set of Munc13-1 $C_2A$ domain mutant variants generated (Fig. 3a). Synapses that expressed the homodimerization-disrupting mutant Munc13-1 (K32E) showed docking activity comparable to Munc13-1 WT rescued synapses (Fig. 4b,c). Munc13-1 (E128K, E137K), which favours the homodimerization state and (K32E, E128K, E137K) which restricts Munc13 to the monomeric state by impairing homo-dimerization and Munc13-1/RIM heterodimerization both showed a 40% reduction in docked SVs (Fig. 4b,c). No groups showed changes in the number of SVs within 100 nm from the AZ, in SV diameter or in the length of the AZ (Fig. 4d and Supplementary Table 1). These results are consistent with the impairment of vesicle docking observed with the Munc13-1 truncation mutants (Fig. 1b,c), and more importantly demonstrate that the $C_2A$ domain-dependent modulation of vesicle docking is not due to disruption of Munc13 homodimerization, but depend on the ability of Munc13-1 and RIM to heterodimerize.

**Heterodimeric Munc13 favours vesicle priming and release.** We next compared vesicle priming and neurotransmitter release characteristics in autaptic hippocampal neurons. Rescue of *Munc13-1/2* DKO neurons with the homodimerization-disrupting mutant Munc13-1 (K32E) showed full rescue of vesicle priming, demonstrated by the measurement of the RRP size (Fig. 5a,b), as well as for evoked release (Fig. 5c,d). Expression of this mutant tended to increase the $P_{vr}$, although the difference compared to WT was not significant, suggesting that the Munc13/RIM binding does not affect the fusogenic state of the vesicles (Fig. 5e). In contrast, Munc13-1 (E128K, E137K) which favors Munc13 $C_2A$ domain homodimerization, strongly impaired vesicle priming and AP-evoked responses. Expression of the triple point mutant Munc13-1 (K32E, E128K, E137K), which arrests Munc13-1 in its monomeric state led to 50% reduced RRP size and AP-evoked EPSC amplitudes without affecting $P_{vr}$ (Fig. 5a–e). Paired-pulse experiments showed that the second EPSC amplitude was slightly facilitated in synapses expressing the Munc13-1 (E128K, E137K) mutant which favours homo-dimerization (Fig. 5f,g), but to a lesser extent than the $C_2A$ truncation mutant (Fig. 2f,g). The homodimerization-disrupting mutant Munc13-1 (K32E) showed mild depression compared with the WT (Fig. 5e) that did not reach significant difference. The paired-pulse ratios of responses from synapses expressing the monomeric Munc13-1 (K32E, E128K, E137K) did not differ from those of the WT.

Together, our data show that Munc13-1 function in priming and NT release is highly dependent on the interactions of its $C_2A$ domain. The Munc13-1 (E128K, E137K) mutant, which is unable

---

**Figure 3 | Point mutants in Munc13-1 $C_2A$ domain result in homodimeric and monomeric Munc13-1 and heterodimeric complex of Munc13-1 and RIM.** (**a**) Schematic representation of the predicted states of monomerization, homodimerization and heterodimerization resulting from the introduction of the single point mutation (K32E), the double point mutation (E128K, E137K) and the triple point mutation (K32E, E128K, E137K) within the $C_2A$ domain of Munc13-1. (**b**) Ribbon diagrams of the Munc13-1/RIM2α heterodimer with the Munc13-1 fragment containing the $C_2A$ domain (residues 3–150) in orange and the RIM ZF domain (residues 82–142) in blue. Zinc atoms are shown as yellow spheres. The diagram in the top panel shows the overall structure of the complex. The bottom left and right panels show the heterodimerization interface of the RIM2α ZF domain with the Munc13-1 $C_2A$ domain and C-terminal α-helix, respectively. The mutated side chains are labelled. (**c**) ITC analysis of binding of WT (left) and double point mutant E128K/E137K (right) Munc13-1 $C_2A$ domain (residues 3–209) to the RIM ZF domain (residues 82–142). (**d**) Co-immunoprecipitation (CoIP) of Munc13-1-Flag in HEK293 cells transiently double transfected with Munc13-1-Flag and WT or mutants Munc13-1 GFP tagged as indicated in the blot (top). (**e**) CoIP of RIM-1α in HEK293 cells transiently double transfected with RIM-1α-RFP and Munc13-1 WT or mutants GFP tagged as indicated in the top blot (bottom). These results are representative of three independent experiments. Molecular weights (kDa) are indicated on the left side and antibodies used for detection on the right side of the blots. The antibody used for the pull down is shown on the top. (**f**) Maximum projection confocal images showing the presynaptic localization of WT, single point mutation (K32E), double point mutation (E128K, E137K) and triple point mutation (K32E, E128K, E137K) Munc13-1 expressed in *Munc13-1/2* DKO neurons. DKO hippocampal neurons were fixed at 16 days *in vitro* (DIV) and counterstained with GFP and VGLUT1 antibodies. Scale bar, 10 μm.

to interact with RIM to disassemble the $C_2A/C_2A$ homodimers is deficient in vesicle priming and exhibit a tendency to facilitate (Fig. 5h,i). In contrast, the Munc13-1 (K32E) mutant, which promotes Munc13-$C_2A$/ZF-RIM heterodimerization shows full priming activity and evoked release. The direct comparison of the data from the monomeric Munc13-1 (K32E, E128K, E137K) with the heterodimeric Munc13-1 (K32E) mutant demonstrate that Munc13 monomers are less efficient in priming and neurotransmitter release compared with Munc13-1/RIM heterodimers (Fig. 5h,i).

**$C_2A$ domain regulation acts upstream of $C_1$ domain activation.** Our previous electrophysiological analysis of the Munc13-1 point mutants that promote heterodimerization revealed a novel regulation of docking via the Munc13 $C_2A$ domain and a similar effect on vesicular priming as that described by Deng et al.[37] It is known that the activation of the Munc13-1 $C_1$ domain potentiates vesicular release probability by increasing vesicle fusogenicity[29]. To determine whether the observed modulation of the $C_1$ domain is still present in the $C_2A$ point mutants, we tested how application of phorbol 12,13-dibutyrate (PDBu) affects the regulation by the $C_2A$ domain.

We first probed putative docking effects by exposing Munc13-1/2 DKO neurons and DKO neurons rescued with Munc13-1 WT or Munc13-1 $C_2A$ point mutants to PDBu. The number of docked synaptic vesicles was quantified within each group and compared with each corresponding PDBu-untreated group. PDBu treatment did not cause a change in the number of docked SVs in Munc13-1/2 DKO. The same effect was observed on DKO synapses rescued with Munc13-1 WT or the Munc13-1 $C_2A$ homodimerization- and heterodimerization-disrupting mutants (Fig. 6a,b), suggesting that DAG/phorbol ester binding to the Munc13-1 $C_1$ domain does not affect the docking process. Moreover, using electrophysiological recordings we assayed vesicular priming in the DKO neurons expressing Munc13-1 WT and $C_2A$ point mutants before and after treatment with PDBu. We observed that the presence of PDBu did not change the RRP size in any of the group tested (Fig. 6c). The lack of effect of PDBu on vesicle docking and priming in all the groups is consistent with a post-priming role of DAG/phorbol ester binding to the Munc13-1 $C_1$ domain[29].

We finally examined the effect of PDBu treatment on AP-evoked neurotransmitter release and $P_{vr}$ of neurons expressing the $C_2A$ homodimerization- and heterodimerization-disrupting mutants. PDBu application rapidly potentiated AP-evoked EPSC amplitudes in all the mutants. By comparing the degree of potentiation, we observed stronger PDBu potentiation in the $C_2A$ mutants tending towards lower initial $P_{vr}$, and less potentiation in the $C_2A$ homodimer-disrupting mutant Munc13-1 (K32E), which tend to have high initial $P_{vr}$ (Fig. 6e). Importantly, the absolute $P_{vr}$ reached similar levels in the presence of PDBu across all mutants (Fig. 6f). These observations indicate that the activation of the $C_1$ domain by DAG/phorbol ester binding normalizes any regulation of the $C_2A$ domain on release probability and defines the $C_1$ domain regulation as downstream of the $C_2A$ domain activity.

**Discussion**
Synaptic transmission efficiency is critical for enabling precise information flow for all brain functions. Munc13s act as master regulators of various aspects of neurotransmitter release through the multiple domains that define its long protein sequence. The identification of an interaction between the RIM N-terminal ZF domain and the Munc13 $C_2A$ domain[35,39], together with the interaction of Rab3 with two α-helices adjacent to the RIM ZF

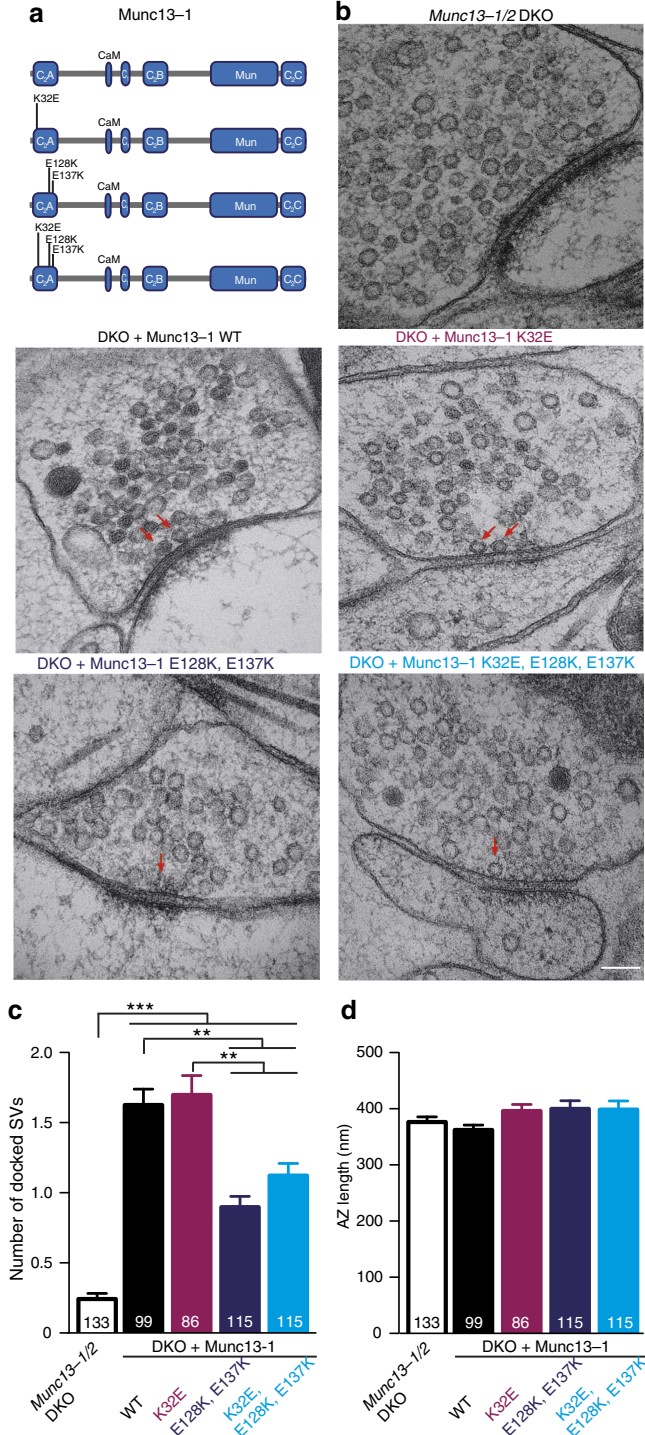

**Figure 4 | The Munc13-1/RIM heterodimer complex is optimal for docking.** (**a**) Scheme corresponding to the structure of the full-length Munc13-1 (WT), homodimerization mutant (K32E), heterodimerization mutant (E128K, E137K) and homo/heterodimerization mutant (K32E, E128K, E137K) used for rescues in Munc13-1/2 DKO hippocampal neurons. (**b**) Representative electron micrographs of synapses showing docked synaptic vesicles (indicated by red arrows) from Munc13-1/2 DKO hippocampal cultures and DKO rescued with the respective Munc13-1 WT and mutants indicated above. Scale bar, 100 nm. (**c**) Plot of number of docked SVs. (**d**) Plot of AZ length in nm. Numbers in bar graphs represent the n values for each group. Significances and P values were determined by one-way analysis of variance (ANOVA) with Kruskal–Wallis test followed by Dunn's post test. Values indicate mean ± s.e.m.; $**P < 0.01$; $***P < 0.001$.

domain[42], suggested that the binary Munc13–RIM complex or a ternary complex with involvement of Rab3 could play important roles in docking and/or priming of SVs. However, the finding that most of the priming defect observed in the absence of αRIMs can be rescued by a N-terminally truncated ubMunc13-2 or by a

ubMunc13-2 with a point mutant at the $C_2A$ domain that impairs homodimerization[37], led to the proposal that the Munc13 $C_2A$ domain acts solely as a switch. When two Munc13 $C_2A$ domains homodimerize, vesicle priming and release is blocked, and disruption of the homodimer by RIM binding activates release.

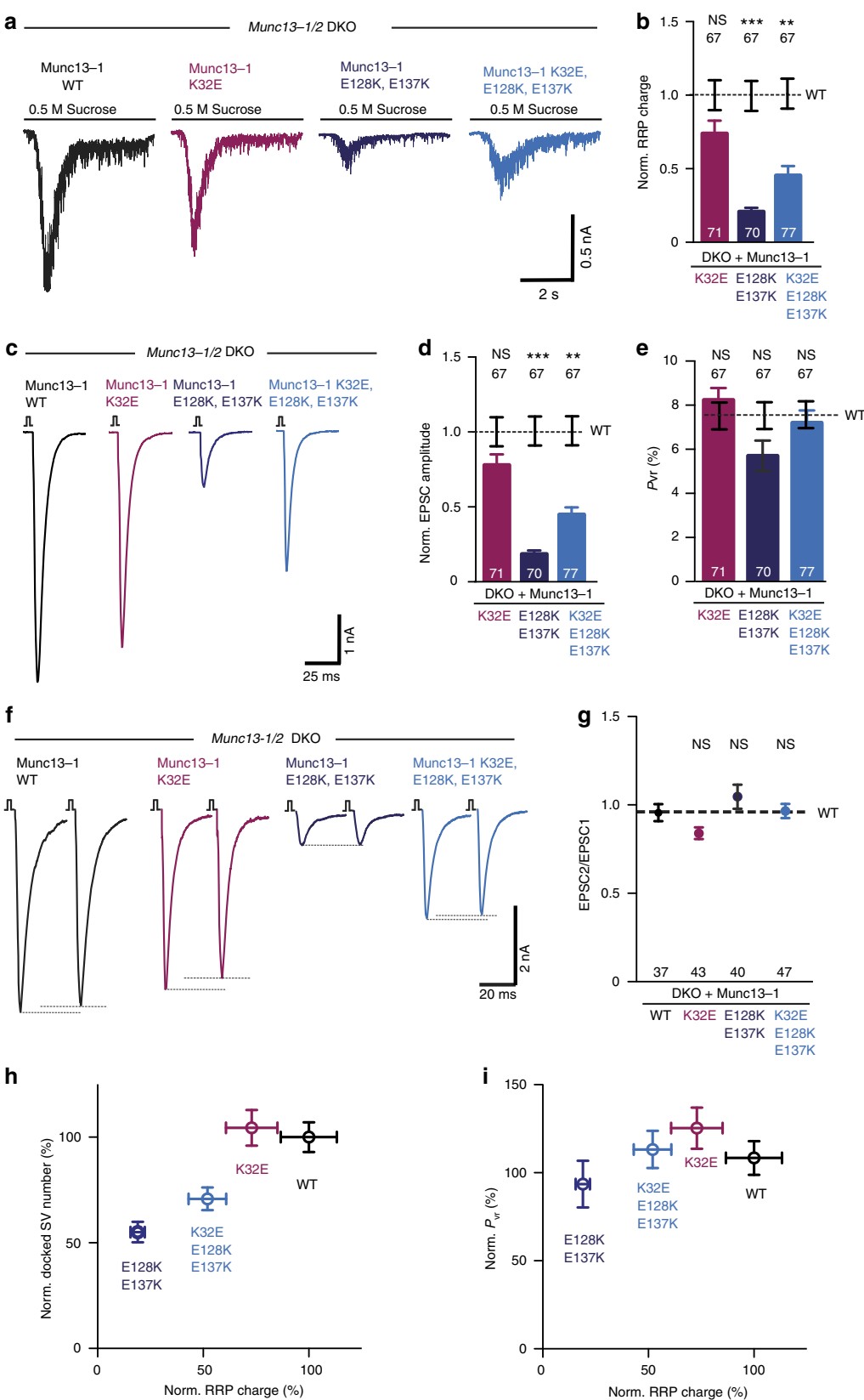

Our study supports the proposal that disruption of the Munc13 homodimer by RIM is indeed critical for its function, but we also show that the Munc13-1 $C_2A$ domain does not act merely because a switch, because heterodimerization of the $C_2A$ domain with RIM optimizes Munc13 function. Thus, by forming a complex with RIM, the Munc13-1 $C_2A$ domain plays a number of

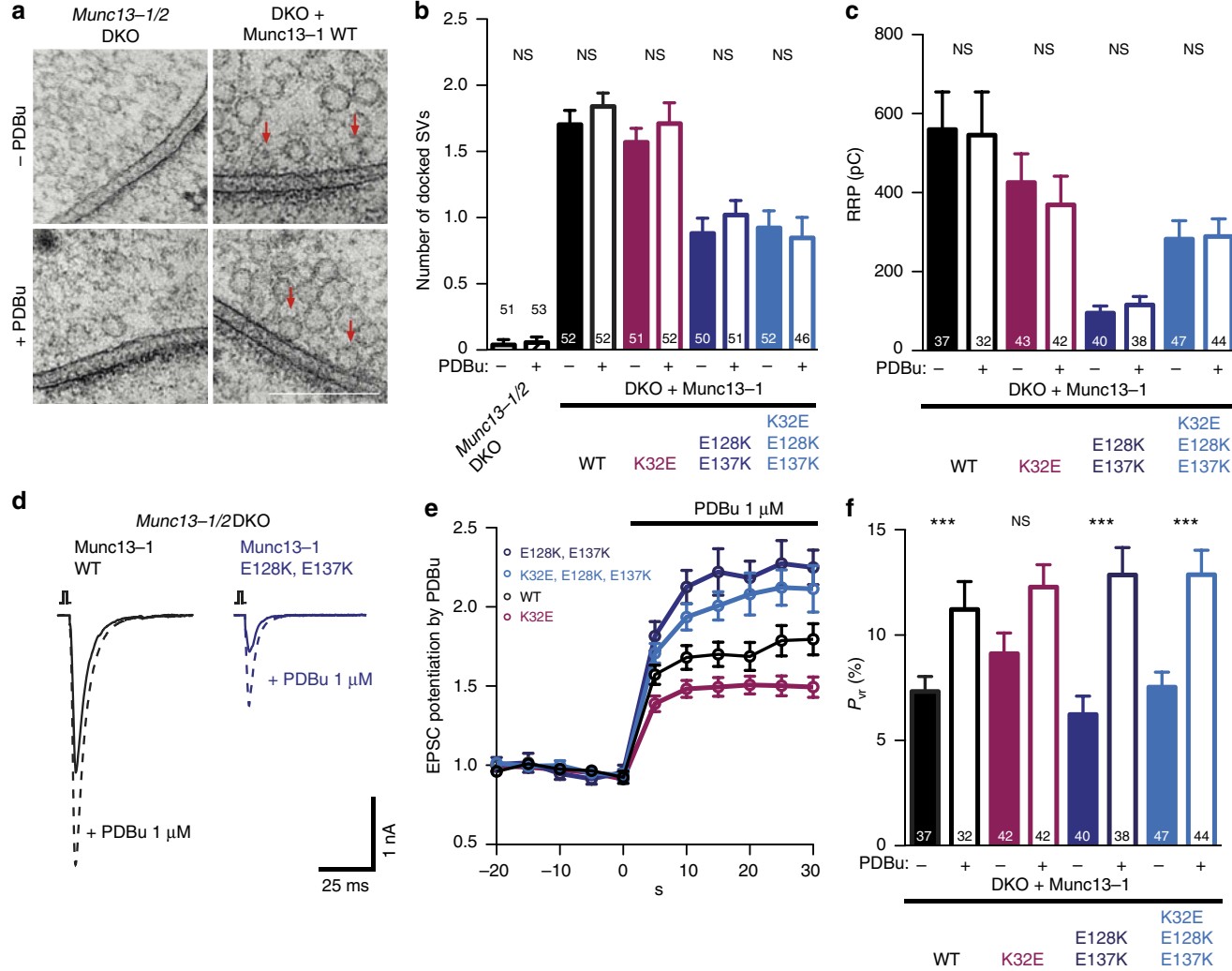

**Figure 6 | Activation of the $C_1$ domain by DAG/phorbol ester is downstream of the regulation of $C_2A$ domain.** (**a**) Representative electron micrographs of synapses showing docked synaptic vesicles (indicated by red arrows) from *Munc13-1/2* DKO synapses and DKO rescued with Munc13-1 WT. Scale bar, 200 nm. (**b**) Plot of docked synaptic vesicles from the *Munc13-1/2* DKO and DKO synapses rescued with Munc13-1 WT and Munc13-1 $C_2A$ homodimerization- and heterodimerization-disrupting mutants with or without PDBu. (**c**) Plot of RRP charge of DKO neurons rescued with Munc13-1 WT and Munc13-1 $C_2A$ homodimerization- and heterodimerization-disrupting mutants with or without PDBu. (**d**) Example traces of evoked EPSC amplitudes from *Munc13-1/2* DKO rescued with Munc13-1 WT in black and Munc13-1 that favours the homodimerization state E128K, E137K in blue (solid lines) and their corresponding EPSCs after PDBu application (dotted lines). (**e**) Potentiation of AP-evoked EPSC amplitudes induced by 1 μM of PDBu. PDBu amplitudes were calculated by normalizing the EPSC amplitude in PDBu with the preceding EPSCs recorded in control extracellular solution. (**f**) Vesicular release probability $P_{vr}$ for Munc13-1 WT and mutant rescues with or without PDBu. Numbers in bar graphs are *n* values for each group. Data are expressed as mean ± s.e.m. For each mutant group, significance and *P* values were calculated by comparison with the non-PDBu-treated group using the unpaired Student's *t*-test: Mann–Whitney. \*\*\*$P < 0.001$.

**Figure 5 | Munc13-1/RIM heterodimerization is required for optimal vesicle priming.** (**a**) Representative traces of RRP charges induced by 0.5 M sucrose from DKO rescued with Munc13-1 WT and point mutants indicated above. (**b**) Plot of RRP charge of the Munc13-1 mutant rescues normalized to WT Munc13-1. (**c**) Representative traces of evoked EPSC amplitudes from *Munc13-1/2* DKO rescues with Munc13-1 WT in black, Munc13-1 K32E in purple, Munc13-1 E128K, E137K in dark blue and Munc13-1 K32E, E128K, E137K in light blue. (**d**) Plot of AP-evoked EPSC amplitudes of Munc13-1 mutant rescues normalized to the corresponding Munc13-1 WT. (**e**) Calculated vesicular release probability $P_{vr}$ in %. (**f**) Example traces of EPSC amplitudes in responses to 2 APs separated by 100 ms (10 Hz) from DKO rescues with Munc13-1 WT and mutants indicated above. (**g**) Graph showing average paired-pulse ratios calculated from the 2 AP-evoked EPSC amplitudes. (**h,i**) Correlation between docked synaptic vesicles, primed synaptic vesicles and vesicular release probability from DKO neurons rescued with Munc13-1 WT and Munc13-1 $C_2A$ homodimerization- and heterodimerization-disrupting mutants. Numbers in bar graphs are *n* values for each group. Error bars represent s.e.m. For each mutant, group significance and *P* values for (**b,d,e,g**) were calculated by Kruskal–Wallis one-way analysis of variance followed by a multiple comparison Dunn's *post hoc* test. Values indicate mean ± s.e.m.; \*\*$P < 0.01$; \*\*\*$P < 0.001$.

important roles in SV docking, priming and neurotransmitter release.

That Munc13 is indispensable in the mammalian vesicle docking process has been demonstrated by recent 3D electron tomography and TEM studies on high-pressure frozen Munc13-deficient synapses[3–5]. The complete loss of docked synaptic vesicles is in agreement with results obtained in earlier studies using *C. elegans* *unc13*-null mutants[2,17,18]. However, the mechanism of Munc13-mediated docking remains to be determined. Our study shows that efficient SV docking and priming are closely linked to the persistent formation of Munc13/RIM heterodimers. Due to the fact that all the mutants showed synaptic responses and were localized at VGLUT1-positive compartments, we suggest that Munc13 can co-exist as a homodimer, a monomer and heterodimer with RIM in the presynaptic terminal (Fig. 3a). The equilibrium between these three states depends on the strength of the interactions with either another Munc13 $C_2A$ domain or with RIM. We utilized truncations of the Munc13-1 $C_2A$ or the N terminus (Fig. 1), as well as three Munc13-1 $C_2A$ domain point mutants that disrupt homodimerization and/or Munc13-1/RIM heterodimerization (Fig. 4), and show that docking is fully rescued in synapses expressing mutants that support heterodimerization. Conversely, docking is reduced in synapses expressing the mutants that lack the Munc13-1 $C_2A$ domain or disrupt Munc13-1 $C_2A$–RIM interaction (Figs 1 and 4). These observations not only suggest that RIM activates Munc13 function by dissociating the Munc13 dimers,[37] but also the Munc13/RIM heterodimer complex is required for full SV docking activity. Interestingly, SV docking is only compromised by 40–50% when the interaction of the Munc13-1 $C_2A$ domain with RIM is disrupted, in contrast to the complete loss of vesicle docking in the absence of Munc13 (refs 3–5). Since deletion of the $C_2A$ domain or point mutants that impair the RIM binding affect docking activity, but show difference in expression levels, suggests that changes in Munc13 expression in the presynaptic compartment are unlikely to be the main cause of the effect seen in vesicle docking activity. However, Munc13/RIM interaction may be essential for the docking process and the 50% docking activity observed in the Munc13-1 (del 1–150), Munc13-1 (del 1–520), Munc13-1 (E128K, E137K) and Munc13-1 (K32E, E128K, E137K) mutants, are the result of a local reduction of Munc13 translocation at the AZ, given the proposed role of RIM in recruitment of, Munc13 to the active zone[36]. In fact, all constructs that prevented Munc13-1/RIM heterodimers had a clear diffused axonal or dendritic signal compared with the WT (Supplementary Figs 1 and Fig. 3f). Possibly Munc13 content is reduced directly at the AZ that we are however not able to assess with the resolution of our imaging method. Other alternatives that could explain the remaining docking activity include, the existence of other RIM/Munc13 interactions that are not mediated by the Munc13-1$C_2A$/RIM-ZF domains. Indeed, RIM protein was still detected in the co-immunoprecipitation assays that utilized the mutant described above (Fig. 3 and Supplementary Figs 3 and 5). Or that disruption of Munc13/RIM/Rab3 tripartite complex impairs vesicle docking[11]. Finally, the primary docking role of Munc13 may be mediated via its C terminus by the membrane bridging activity proposed by Liu *et al.*[26] that may favour efficient formation of trans-SNARE complexes[4,24,25] and be enhanced and regulated by the $C_2A$ domain RIM interaction at the N terminus.

SV docking precedes priming and is therefore a prerequisite for rendering fusion competent vesicles. Accordingly, we found that all Munc13 truncations and point mutants that impaired vesicle docking (Figs 1c and 4c) also had a vesicle priming defect (Figs 2b and 5b), but the fact that the decreased SV docking did not correspond to the same extent with the decrease in SV priming in the case of the $C_2A$ deletion and homodimerization point mutants suggest that docking is not necessarily equivalent to

vesicle priming (Figs 2h and 5h). In addition, our functional analysis indicates that the formation of the Munc13-1 $C_2A$/ZF RIM heterodimer complex is the optimal configuration for vesicle priming and neurotransmitter release. The importance of the heterodimerization of the $C_2A$ domain with RIM was apparent when comparing the release properties of the single point mutant (K32E), the Munc13-1 monomer that can bind to RIM, with the triple point mutant (K32E, E128K, E137K), the Munc13-1 monomer that lacks RIM binding activity (Fig. 5e). The underlying mechanism for the boosted function of the Munc13/RIM heterodimer is unclear, but may require the presence of Rab3 that can form a tripartite complex together with Munc13-1 and RIM[35]. Rab3-deficient synapses have been found to have reduced vesicular release probability[38], supporting the hypothesis that the tripartite complex may have a boosting function on release. Finally, we found that the activity of Munc13-1/RIM complex is distinct and upstream of the potentiation of vesicle release probability induced by the binding of DAG/phorbol ester to the $C_1$ domain (Fig. 6) that is thought to contribute to short-term plasticity events[27,43,44]. Thus, following basal vesicle docking and basal priming, the vesicular release can be positively modulated by membrane translocation driven by the DAG/$C_1$ domain interaction.

Our structure–function analysis also provides further insights into how the N terminus of Munc13 contributes to the regulation of vesicle fusion efficiency. We found that the intermediate region between the $C_2A$ and the $C_1$ domains (residues 150–519) is a novel negative regulator of vesicular release probability, as deletion of residues 1–150 impairs release more strongly than deletion of residues 1–520 (Fig. 2). This regulatory function is probably not related to CaM binding, as disruption of the CaM-binding activity does not change vesicular release probability and instead causes accelerated depression during AP trains[32,45]. Further structure–function analysis of the Munc13 N-terminal region will be required to gain insights into the inhibitory function. Nevertheless, the discovery of the inhibitory activity in the intermediate region between the $C_2A$ and $C_1$ domains help explain the observation that deletion of the entire N-terminal region leads to small changes in vesicle priming and release[37]. Loss of release through the lack of the $C_2A$ domain together with the disinhibition from the deletion of the N-terminus intermediate region may prevent a net change in overall synaptic output (Fig. 2).

Overall, our results revise the previous interpretation that the $C_2A$ domain of Munc13 and the ZF domain of RIM protein are dispensable for vesicle priming once Munc13 homodimers are disrupted[37]. Our findings emphasize the positive impact of the $C_2A$ domain of Munc13 to enable this protein to orchestrate, through its dynamic interactions with RIM, a wide range of modulatory operations to shape vesicle docking, priming and fusion.

## Methods

**Munc13-1 rescues lentiviral constructs and virus production.** Munc13-1 full-length, truncation, deletions and point mutants rescue construct were generated from a rat *Unc13a* splice variant, similar to *Unc13a* (NM_022861) but lacking the exons 48, 49 and 53. The complementary DNAs (cDNAs) were subsequently cloned into a lentiviral shuttle vector under the control of a human *synapsin-1* promoter. To enable identification of infected cells, Munc13-1 rescue proteins were fused with fluorescent protein tagged Venus. Concentrated lentiviral particles were prepared as previously described[46].

**Autaptic hippocampal culture and lentiviral infection.** *Munc13-1/2* DKO mice on a FVB/N background were used for all studies. Animal Welfare Committee of Charité—Universitätsmedizin and the Berlin state government agency for Health and Social Services approved all protocols for animal maintenance and experiments (license no. T 0220/09). Microislands of growth-permissive substrate (collagen and poly-D-lysine) were printed onto 30 mm round glass coverslips precoated with

agarose type 4, using a stamp containing $200 \times 200 \mu m$ spaced dots. Astrocytes were plated at a density of 5,000 cells per $cm^2$ onto the microdot-coated coverslips to allow them to grow onto the growth-permissive substrate. Hippocampi were dissected from embryonic day 18.5 Munc13-1/2 DKO mouse and enzymatically treated with 25 units per ml of papain for 45 min at 37 °C. After enzyme digestion, solution was exchanged for an inactivation solution containing albumin, trypsin inhibitor and 5% fetal calf serum. Hippocampi were mechanically dissociated and the neuron suspension was plated onto the astrocyte microislands at a final density of 300 cells per $cm^2$. Neurons were incubated at 37 °C for 13–16 days to mature before initiating the experiments. For rescue experiments neurons were infected 24 h after plating with lentiviral vectors.

**High-pressure freezing and EM.** Munc13-1/2 DKO or DKO hippocampal neurons infected with the individual rescue constructs were plated at a density of 50,000–100,000 cells onto 6 mm sapphire disks for 13–16 days. Neuronal cultures were frozen using a high-pressure freezer (HPM 100 Leica) under a pressure of 2,100 bar by mounting it into a sandwich support with extracellular solution containing 15% Ficoll. The cryofixation was achieved within milliseconds allowing simultaneous immobilization of all macromolecular components. After freezing, samples were transferred into cryovials containing 1% glutaraldehyde, 1% osmium tetroxide, 1% ddH2O in anhydrous acetone and processed in an AFS2 automated freeze-substitution device (Leica) with the following temperature program: 5 h at − 90 °C, from − 90 °C to − 20 °C (5 °C per h), 12 h at − 20 °C and from − 20 °C to 20 °C (10 °C per h). After freeze substitution, samples were en bloc stained in 0.1% uranyl acetate for 1 h and infiltrated in increasing concentration of epoxy resin (Epon 812, EMS) diluted in acetone at room temperature. Subsequently, samples were flat embedded in Epon and baked for 48 h at 60 °C to allow Epon polymerization. Approximately 200–300 μm square areas containing neurons were randomly selected and sectioned at 40–50 nm thickness using an Ultracut UCT ultramicrotome (Leica) equipped with a diamond knife (Diatome ultra 45). Ultrathin sections were collected on formvar-coated (0.5%) 200 mesh copper grids (EMS) and counterstained with 2% uranyl acetate and lead citrate (0.15 m lead nitrate, 0.12 m sodium citrate in ddH2O). Sections were examined in a FEI Tecnai G20 TEM operated at 80–120 keV and digital images were taken with a Veleta 2 K × 2 K CCD camera (Olympus). EMs (2,048 × 2,048 pixels) containing a single synaptic profile were collected blindly at $135,000 \times$ magnification (pixel size 0.38 nm). Data were analysed blindly by using an analysis program developed for ImageJ and Matlab as reported by Watanabe et al.[5] Active zones were defined as the portion of the presynaptic membrane directly opposite to the postsynaptic density. Docked synaptic vesicles were defined as vesicles observed to be in direct contact with the plasma membrane, that is, in the images the contact between vesicle and plasma membrane was gapless. Three independent Munc13-1/2 DKO hippocampal neuronal cultures were used for each data set.

**Electrophysiology.** Synaptic function was assayed by whole-cell voltage clamp from autaptic hippocampal Munc13-1/2 DKO neurons at 13–16 days in vitro. Synaptic currents were monitored at room temperature using a Multiclamp 700B amplifier (Axon Instrument). The series resistance was compensated by 70% and only cells with series resistances of $< 10 M\Omega$ were analysed. Data were acquired using pClamp 10 software (Axon Instrument) at 10 kHz and filtered using a low-pass Bessel filter at 3 kHz. Borosilicate glass pipettes with a resistance between 2 and 3.5 $M\Omega$ were used and filled with a solution containing the following (in mM): 136 KCl, 17.8 HEPES, 1 EGTA, 4.6 $MgCl_2$, 4 $Na_2ATP$, 0.3 $Na_2GTP$ and 12 creatine phosphate; 50 U $ml^{-1}$ phosphocreatine kinase; adjusted to pH 7.4 with osmolarity at ∼300 mOsm. Neurons were continuously perfused with standard extracellular solution including the following (in mM): 140 NaCl, 2.4 KCl, 10 HEPES, 10 glucose, 2 $CaCl_2$ and 4 $MgCl_2$; adjusted to pH 7.4 with osmolarity at ∼300 mOsm. AP-evoked EPSCs were triggered by 2 ms somatic depolarization from − 70 to 0 mV. To determine the size of the RRP, 500 mM sucrose was applied directly onto the isolated neuron for 5 s using a fast-flow system[47]. A transient inward current lasting for 2–3 s that characterizes the release of the RRP is followed by a steady-state current that indicates the point at which the rates of refilling and release are equal. When the steady-state current is used as a baseline, the area under the transient curve indicates the total charge released by the RRP . For vesicular release probability ($P_{vr}$ ) calculations, the ratio of EPSC charge to RRP charge was determined. Paired-pulse stimulation protocol contained two inductions of an AP at an interval of 100 ms (10 Hz). The paired-pulse ratio was calculated by dividing the amplitude of the second EPSC by amplitude of the first. For phorbol ester experiments, we first determined the RRP size under resting condition. One minute later, EPSC amplitudes were monitored at 0.2 Hz for 30 s. without PDBu application and 30 s under application of 1 μM of PDBu to determine the degree of EPSC potentiation. Immediately after the application of PDBu, the RRP size was again measured. Data were analysed offline using Axograph X (Axograph Scientific).

**Isothermal titration calorimetry.** ITC experiments were performed on a VP-ITC system (MicroCal) at 20 °C in a buffer containing 50 mM Tris (pH 7.3), 150 mM NaCl, 5 mM $MgCl_2$ and 1 mM TCEP as described[35]. Data were fitted to a single binding site model using a nonlinear least-squares routine within Origin for ITC v.5.0 (MicroCal).

**Transfection, coimmunoprecipitation and western blotting.** HEK293 expression vectors for Munc13-1 WT and $C_2A$ domain mutants were constructed in pEGFP-N1 (CLONTECH Laboratories). Transgenes were cloned between the cytomegalovirus promoter and the SV40 polyA sequences. Munc13-1-FLAG was generated from rat Unc13a by PCR amplification with a reverse primer that harbours a 3xFLAG sequence (Sigma-Aldrich). The RIM-1α cDNA was a gift from Dr Thomas C Südhof at the Stanford University and RFP sequence was fused to the C terminus.

HEK293TN (BioCat GmbH) cells were transiently double transfected at 80% confluency with: (1) Munc13-1 with 3xFLAG fused to the C terminus and Munc13-1 WT or $C_2A$ mutants plasmids with C-terminal GFP tag to detect $C_2A$–$C_2A$ interaction or (2) RIM-1α with RFP fused to the C terminus and C-terminally GFP-tagged Munc13-1 WT or $C_2A$ domain mutants to assess RIM/Munc13 interaction. After 48 h, cells were washed twice with phosphate-buffered saline (PBS) and lysed at 4 °C with a NP-40 cell lysis buffer containing: 50 mM Tris-HCl (pH 7.9), 150 mM NaCl, 1% NP-40, 0.5% deoxycholate and 5 mM EDTA and protease inhibitor cocktail-complete mini.

Next, 250 μg of the double transfected lysates were incubated for 30 min at room temperature with protein G-Dynabeads (Thermo Scientific, Hampton, NH, USA) bound to the antibody (Ab) Rabbit anti-GFP (Abcam 6556) or control Ab Rabbit IgG (Cell Signaling 2729S). The Dynabeads-Ab-Antigene (Ag) c were washed three times with washing buffer and eluted with 25 μl elution buffer. Final 25 μl eluted Dynabeads-Ab-Ag complexes were mixed with 25 μl sample buffer (Laemmli 2 × concentrated containing 0.2 M dithiothreitol) and boiled for 10 min at 70 °C. Dynabeads were removed after boiling and 25 μl of the immunoprecipitates and 25 μg protein from the double transfected lysates were used for the SDS–polyacrylamide gel electrophoresis. Proteins were transferred to nitrocellulose membranes. After blocking with 5% skim milk in Tris-buffered saline with Tween-20, blots were incubated with mouse monoclonal ANTI-FLAG M2 (Sigma F1804), Living Colors A.v. monoclonal Ab (JL-8) (Clontech 632380) or rabbit polyclonal anti RIM1/2 (SYSY 140203) Abs. After washing the blots with Tris-buffered saline with Tween-20, the immune reactive proteins were detected by ECL (ECL-plus kit; GE Healthcare Biosciences) with horseradish peroxidase-coupled secondary Ab (Jackson ImmunoResearch Laboratories). Western blot images (Fig. 3d) have been cropped for presentation. Full size images are presented in Supplementary Figs 3 and 5.

**Immunocytochemistry.** Munc13-1/2 DKO or DKO hippocampal neurons infected with the different rescue constructs were plated at a density of 25,000 cells and fixed in 4% paraformaldehyde in PBS at 16 days in vitro. After fixation, neurons were permeabilized with PBS–Tween 20, blocked with serum in PBS–Tween 20 and incubated overnight with rabbit polyclonal antibody against GFP (1:500; Abcam 6556) and guinea pig polyclonal antibody VGLUT1 (1:4,000; Synaptic System, 135304). Primary antibodies were labelled with Alexa Fluor 488 Affinipure donkey anti-rabbit IgG and Alexa Fluor 647 Affinipure donkey anti-guinea pig IgG (1:500 each; Jackson ImmunoResearch). Coverslips were mounted with Mowiol 4–88 antifade medium (Polysciences Europe). Fixed neurons were imaged on a confocal laser-scanning microscope Leica TCS SP8 with identical settings used for all samples. Neuronal cultures were visualized using a 63 × oil immersion objective. Images were acquired using Leica Application Suite X (LAsX) software at 1,024 × 1,024 pixels resolution using a z-series projection of 8–10 images with 0.3 μm depth intervals. Ten independent neurons per group for each cultured and three different cultures were imaged and analysed using ImageJ software.

**Statistics.** For the statistical analysis data were collected from at least three independent hippocampal cultures to minimize culture-to-culture variability. To minimize variability among the electrophysiological data sets, an approximately equal number of autaptic neurons were recorded from all experimental groups per day. Electrophysiological data from the different experimental groups during 3–4 consecutive days of recording were normalized to the mean value of the control group. Data are expressed as mean ± s.e.m. Statistical comparison was performed in all figures except for Fig. 6 by Kruskal–Wallis one-way analysis of variance followed by a multiple comparison Dunn's post hoc test. For the data in Fig. 6 (PDBu experiment), significance and P values were calculated by comparison with the non-PDBu-treated group using the unpaired Student's t-test: Mann–Whitney. Data were acquired and analysed in a blinded fashion and differences between data sets were considered significant at $P < 0.05$.

**Data availability.** The data that support the findings of this study are available from the corresponding author on reasonable request.

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

## Acknowledgements

This research was supported by the German Research Council grant SFB958 (to C.R. and M.C.), the ERC grant SynVglut (to C.R.), the National Institutes of Health grant R35 NS097333 (to J.R.) and the Spanish Foundation of Manuel Morales and the Spanish Ministry of Science and Innovation (to M.C.). We thank Rike Dannenberg, Annegret Felies, Berit Söhl-Kielczynski, Sabine Lenz, Bettina Brokowski and Katja Pötschke for technical assistance, and the Charite viral core facility for virus production and characterization. We thank Dr Melissa Herman and Dr Benjamin Rost for fruitful discussions and comments on the manuscript and Dr Tanja Rosenmund for assistance with animal care.

## Author contributions

C.R., M.C., J.B., J.R. and I.D. conceived and designed experiments. M.C., S.C., S.-S.C., M.A.-R. and C.P.-L. conducted experiments. T.T. provided molecular reagents. M.C., C.R. and J.R. wrote and edited paper.

## Additional information

**Competing interests:** The authors declare no competing financial interests.

