## [Peer Review File · Nature Communications]

Reviewers' comments:

Reviewer #1 (Remarks to the Author):

The manuscript dissects roles of the Munc13 N-terminal sequences in synaptic vesicle docking (measured by electronmicroscopy on high pressure frozen samples, but without tomography), synaptic vesicle release (measured as the EPSC amplitude), RRP (measured as vesicles released by hypertonic sucrose) and Pvr (measured qualitatively as PPR, calculated as EPSC/RRP). The data are of high quality, they extend previous models of Munc13 function and a key strength is the systematic assessment of all parameters for all mutants in the same preparation (autaptic cultures of hippocampal excitatory neurons). The authors find:

1. The C2A domain of Munc13 contributes to vesicle docking and to RRP
2. The linker sequence appears to suppress Pvr
3. RIM binding of the C2A domain, but not homodimerization, is required for its roles in docking and RRP.

The data support a previous model in which the interaction between Munc13 C2A and RIM has an activating function on RRP, and they extend the model by revealing a novel role in vesicle docking. There are additional effects on Pvr, but those are generally small when assessed independently of RRP by paired pulse ratios. Given that the Pvr effects map onto the linker and not the C2A domain, the paper title is factually wrong and should be adjusted.

The data are convincing, the manuscript is well written and the findings significantly extend the understanding of Munc13 and its roles in synaptic vesicle release. I am in support of publishing this manuscript in Nature Communications. However, there are a few points that the authors should address before publication:

Major comments:

1. Interpretation of the docking roles:

The data can be interpreted in two different ways: (i) Munc13 has two separable docking roles, about 50% are mediated by the C2A domain, and another 50% by an unknown mechanism (trans-snare assembly or C2B/C2C mediated functions, see further comments below) or (ii) a single role in docking (via trans-snare assembly or C2B/C2C domains) combined with a known role of RIM in recruiting Munc13 to the active zone (Andrews-Zwilling et al, JBC) such that docking is Munc13 dose-dependent. In the latter scenario, the RIM-binding mutants lead to less Munc13 recruitment to the active zone (which could not be assessed with the resolution of the imaging methods the authors use), and thus to less docking. The authors should at least discuss these two possibilities.

2. Although the data as presented are convincing, the interpretation for some of the mutants is less clear. It is possible that virally expressed WT Munc13 has a gain of function effect on any or all parameters measured. With the current data, it is impossible to judge this, but it would affect all interpretations because all data are normalized to or compared to "WT rescue". It would be preferable that each experiment is compared to a true wild-type control, but I understand that this would require essentially redoing all experiments, which is not feasible in the context of a revision. However, the authors should do a simple experiment in which they use wild-type neurons infected with a control virus or WT Munc13 lentivirus, and they should characterize docking, RRP and Pvr/PPR to determine whether any of these parameters is affected by viral expression of Munc13. This would provide a gage that puts all data in the context of wild-type neurons.

3. A second confound could be the synaptic levels of some of the mutants. For the truncation mutants, no expression/localization data is shown at all. For example, it is possible that the del 1-150 mutant essentially fails to rescue most of the parameters because there are lower levels at synapses. For all

mutants, it should be established whether they are expressed and localize to synapses (at a minimum in a qualitative sense as shown for the point mutants, but preferably the synaptic signal should be quantified).

4. I find the PDBu data preliminary and the least informative part of the manuscript, and they are internally inconsistent: the K32E mutant clearly enhances release upon PDBu as assessed by the EPSC amplitude, but the authors claim that it does not increase RRP or Pvr in response to PDBu. If the enhancement is presynaptic, this must be incorrect, one parameter has to change. The experiment is perhaps confounded by postsynaptic effects, experimental limitations such as receptor saturation, or awakening of silent synapses (this should still enhance the sucrose RRP, though). I think that the most likely explanation is that there is an increase in Pvr, but that the authors fail to detect it due to insignificant trends in the data (division by a non-significantly smaller RRP). Either the authors should do a much better job in characterizing what the effects of PDBu in these mutants are (a better assessment of release probability via measurement of PPR, determination whether PDBu affects localization of rescue constructs differentially, excluding postsynaptic contributions, assessment of the truncations, determination what the contributions of the C1 domain are....) or just remove the data. Given that it is well known that PDBu ultimately acts via the C1 domain, which the authors do not study in this manuscript, it may be better to just do the latter.

Additional points:

5. The statistical analyses for the electrophysiological measurements are incorrect. Rather than T-tests, the authors should do one-way ANOVA (as they do for the EM) followed by post tests which correct for multiple comparisons, and they should make all relevant comparisons.

6. In the methods, it is stated that the resolution of the EM analysis is 2 nm. It is not explained where that number comes from, but the number is similar to the resolution in a paper by Imig et al (Neuron 2014), which used high pressure freezing combined with electron tomography. Given that the authors did not use tomography in the current study, it seems unlikely that they get this same (excellent) resolution, but the resolution is likely lower. This statement should be corrected.

7. In respect to point 5: this is also important because it appears that Munc13 has two different docking roles: one mediated by the C2A domain, and an additional role that, in my view, is either (1) mediated by the MUN domain through SNARE complex assembly (a lot of literature from the Rizo, Rosenmund, Jorgensen labs) or by the other C2 domains (Liu/Rosenmund/Rizo et al, eLife 2016). It is possible that such differential docking roles could be revealed with resolution of 2 nm: for example a tethering function of the C2A domain may produce less tight docking, vs SNARE complex assembly may produce tighter docking. This topic deserves a more in-depth discussion.

8. On the first page of the introduction, it is stated that "Lack of these proteins leads to a complete loss of docked and primed synaptic vesicles, as shown in Munc13-1/2 double deficient (DKO) mammalian neurons^{1,3-6}", The introduction of vesicle docking/tethering in the Munc13 KO deserves a more detailed discussion, as the phenotype strongly depends on the fixation method. In glutaraldehyde fixed samples, no docking deficit is observed (Augustin 1999, Varoqueaux 2002; in contrast to RIM mutants - Han 2011, Kaeser 2011). This is important because the docking role of the C2A domain may operate through RIM. But at the same time the data argue that it is not identical to the RIM phenotype, because if it were, it should be seen in glutaraldehyde. An additional statement at the end of the introduction also relates to this point ("This model is surprising given that RIM, like Munc13, is required for docking^{2,9-11}"). Previously used definitions of docking and effects of Munc13 and RIM deletions in the context of these definitions should be better introduced.

9. The color code in fig 6e is bad, it is not clear which one is WT and which one is K32E.

10. At the bottom of the first page of the introduction the roles for the MUN domain in priming are described. A recent paper (Liu/Rosenmund/Rizo eLife 2016) should be included, which is at odds with an isolated role for the MUN domain in priming.

11. It should be specified where in the RIM cDNA the RFP tag has been added.

Reviewer #2 (Remarks to the Author):

This highly-focused study probes synaptic function as regulated by two proteins, RIM and Munc13. The authors suggest that there is a balance between C2A-mediated Munc13 dimerization and Munc13/RIM heterodimerization. They provide a complex data set on docking, priming, and action potential-induced EPSCs in Munc13 DKO hippocampal cells rescued by two distinct sets of mutants. Firstly, they probe the function of N-terminal domains of Munc13 including C2A. Secondly, they probe Munc13/RIM heterodimerization and the balancing of Munc13 homodimers, Munc13 monomers, and Munc13/RIM heterodimers. The data provided are high quality (but see 3-5 below) but difficulties in convincingly interpreting this data set with respect to mechanisms seems quite daunting.

1. Some of the interpretations are difficult to follow. Compared to Munc13 KO and WT rescue, docking is partly restored (~50%) by any form of Munc13 and yet their discussion states that docking is only intact in synapses expressing mutants that support heterodimer formation. They must be considering intact docking as being the 100% rather than the 50% value. Thus, one wonders why all of the mutant proteins (delta 1-520, EE, and KEE) support partial docking. Does this imply that another docking pathway involving Munc13 uses a mechanism independent of C2A and independent of RIM? The authors do discuss possible alternative pathways but this points out that the data set is difficult to cleanly interpret.

2. Another data set difficult to interpret is that of Fig 5c where putative monomers of Munc13 seem to be as active in rescue as putative Munc13/RIM monomers. This is based on the interpretation of what these mutants represent but the triple mutant would seem to employ a RIM-independent mechanism that is nearly as strong as the RIM-dependent mechanism. There are probably many plausible interpretations involving other protein interactions.

3. In rescue studies, it is important to know whether levels of expression of WT protein is sufficient to fully restore function during the expression period. In the current work, WT and mutant proteins are contrasted but whether WT protein fully restores function is not stated. This should be discussed. Does it matter how much of a protein is expressed during rescue or do the lentiviruses saturate protein expression?

4. In comparing WT and mutant protein expression, it is important to know whether the proteins are expressed at similar levels. In the current work, there is no indication of protein levels for WT or mutant proteins (in hippocampal cells). This may be of particular importance in this work because Munc13 levels are strongly depressed in RIM KO mice. Would failure to interact with RIM enable degradation of various mutant proteins? In Supplement, the authors detect somewhat variable expression of various mutants relative to WT protein in HEK cells. If this were also true in the hippocampal cells, would this affect the phenotypes?

5. In comparing WT and mutant protein expression, it is also important to know whether the proteins localize normally or in a different manner. The authors present Munc13-GFP localization relative to VGLUT1 in Fig. 3 but the results are not discussed except to say that the Munc13-GFP proteins

overlapped with the immunofluorescence signal for VGLUT1 suggesting proper localization. However, it appeared that K32E Munc13 (with or without RIM mutations) localized more broadly than the WT protein. The authors should quantitate co-localizations for these studies to determine whether similar localization is indeed observed, and the results should be discussed.

Reviewer #1:

“There are additional effects on Pvr, but those are generally small when assessed independently of RRP by paired pulse ratios. Given that the Pvr effects map onto the linker and not the C₂A domain, the paper title is factually wrong and should be adjusted.”

The reviewer is correct in stating that our electrophysiological analysis identified a role of the linker region between C₂A and C₁ domains in regulating p_{vr}. However, we also revealed in addition, that the C₂A domain regulates p_{vr} as documented by these two findings: 1. The severe decrease of vesicular release probability and strong facilitation in the paired pulse experiments in the C₂A truncation mutant (del 1-150) (Fig. 2e), (Fig. 2g). 2. the 30% enhanced release probability (Fig. 5e) accompanied by a pronounced depression in paired pulse experiments (Fig. 5g) in the homodimerization-disrupting mutant Munc13-1 (K32E). The increased release probability in the Munc13-1 K32E mutant was also consistent with results obtained with phorbol esters treatment, where the increased release probability was coupled to reduced potentiation of the evoked EPSC (Fig. 6e). Overall, these data indicate that the presence of the C₂A domain is critical for vesicular release probability, and in particular, shows that Munc13/RIM heterodimer formation results in a more efficient Ca-triggered release. We therefore would maintain our statement that the C₂A domain regulates release probability in the title.

1. Interpretation of the docking roles:

The data can be interpreted in two different ways: (i) Munc13 has two separable docking roles, about 50% are mediated by the C₂A domain, and another 50% by an unknown mechanism (trans-snare assembly or C₂B/C₂C mediated functions, see further comments below) or (ii) a single role in docking (via trans-snare assembly or C₂B/C₂C domains) combined with a known role of RIM in recruiting Munc13 to the active zone (Andrews-Zwilling et al, JBC) such that docking is Munc13 dose-dependent. In the latter scenario, the RIM-binding mutants lead to less Munc13 recruitment to the active zone (which could not be assessed with the resolution of the imaging methods the authors use), and thus to less docking. The authors should at least discuss these two possibilities.

We agree with the reviewer that our docking data can be interpreted in different ways. We made major revisions to the relevant paragraph of the discussion in page 21-22 that now include more alternatives.

2. Although the data as presented are convincing, the interpretation for some of the mutants is less clear. It is possible that virally expressed WT Munc13 has a gain of function effect on any or all parameters measured. With the current data, it is impossible to judge this, but it would affect all interpretations because all data

are normalized to or compared to "WT rescue". It would be preferable that each experiment is compared to a true wild-type control, but I understand that this would require essentially redoing all experiments, which is not feasible in the context of a revision. However, the authors should do a simple experiment in which they use wild-type neurons infected with a control virus or WT Munc13 lentivirus, and they should characterize docking, RRP and Pvr/PPR to determine whether any of these parameters is affected by viral expression of Munc13. This would provide a gage that puts all data in the context of wild-type neurons.

We fully agree with the reviewer that putative changes in rescue constructs may lead to alterations in synaptic properties, and it is important to remove possible limitations about the rescue approach. We conducted a detailed analysis that compares vesicle docking, priming, evoked release and release probability between Munc13-1 WT and Munc13-1 KO/rescued with Munc13-1 on the background of Munc13-2 deficiency. These data are now added as Supplementary Fig. 2 to the manuscript and show that the Munc13-1 construct rescues approximately 65-75% of the response seen in Munc13-1^(+/+)/Munc13-2^(-/-) neurons (Supplementary Fig. 2a-c). This small difference between WT control and the rescue could have several reasons, either by the slightly reduced expression level of the viral system used, by the differences in Munc13-1 isoforms present in the Munc13-1^(+/+)/Munc13-2^(-/-) neurons, or by the overall efficiency of the rescue itself. We do think that this maintains the validity of the rescue approach as we compare the behavior of any mutant to the Munc13-1 WT rescue levels. How sensitive do synaptic properties respond to gradual changes in Munc13-1 expression levels? To assess this, we compared synaptic responses of Munc13-1^(+/+)/Munc13-2^(-/-), Munc13-1^(+/-)/Munc13-2^(-/-) and Munc13-1^(+/+)/Munc13-2^(-/-) plus Munc13-1 overexpression and found that neither reducing protein by 50%, in the case of the heterozygous neurons (Fig. 1c *Varoqueaux et al. 2002*) nor mild overexpression of Munc13-1 (Supplementary Fig. 2a) do not lead to significant changes in synaptic properties (Supplementary Fig. 2d). This suggests that synapses are not highly sensitive to overall Munc13-1 protein content variation, in particular given the apparent lack of a gain of function effect of Munc13-1.

3. A second confound could be the synaptic levels of some of the mutants. For the truncation mutants, no expression/localization data is shown at all. For example, it is possible that the del 1-150 mutant essentially fails to rescue most of the parameters because there are lower levels at synapses. For all mutants, it should be established whether they are expressed and localize to synapses (at a minimum in a qualitative sense as shown for the point mutants, but preferably the synaptic signal should be quantified).

In response to the reviewer's rightful criticism, we performed additional immunocytochemistry experiments for the quantification of the expression levels and the localization of the truncated mutants at the synapses. We have described these data now in the result section and added an additional

Supplementary Fig. 1. In summary, we quantified presynaptic Munc13 expression levels using a GFP tag at the C-terminus of the rescue constructs. We first compared overall antiGFP fluorescent signal in VGLUT1 positive compartments between Munc13-1 WT and the truncation mutants rescued. All truncated and deletion mutants were found at presynaptic compartments and show only small differences in expression levels, that did not reach the significance compared with the Munc13-1 WT. However, we do not think that this moderate reduction in expression levels can explain the reduced vesicle docking activity of the (del 1-150) or (del 1-520), because the (del 150-520) mutant fully rescued docking activity, despite its has the same reduced synaptic signal.

4. I find the PDBu data preliminary and the least informative part of the manuscript, and they are internally inconsistent: the K32E mutant clearly enhances release upon PDBu as assessed by the EPSC amplitude, but the authors claim that it does not increase RRP or Pvr in response to PDBu. If the enhancement is presynaptic, this must be incorrect, one parameter has to change. The experiment is perhaps confounded by postsynaptic effects, experimental limitations such as receptor saturation, or awakening of silent synapses (this should still enhance the sucrose RRP, though). I think that the most likely explanation is that there is an increase in Pvr, but that the authors fail to detect it due to insignificant trends in the data (division by a non-significantly smaller RRP). Either the authors should do a much better job in characterizing what the effects of PDBu in these mutants are (a better assessment of release probability via measurement of PPR, determination whether PDBu affects localization of rescue constructs differentially, excluding postsynaptic contributions, assessment of the truncations, determination what the contributions of the C1 domain are....) or just remove the data. Given that it is well known that PDBu ultimately acts via the C1 domain, which the authors do not study in this manuscript, it may be better to just do the latter.

We performed the PDBu experiments to address two major questions: 1. Are effects of phorbol esters via Munc13 C₁ domain activation related to vesicle docking? Initial studies have been proposed that Munc13 recruitment to AZ membrane by phorbol esters causes potentiation of release. Our results are important as they show that phorbol ester modification of release is not associated with vesicle docking or priming, but must be downstream to these processes through regulation of release probability (Basu et al. 2007). 2. Is the regulation of release probability by the C₂A domain related with the effect of C₁ domain? The results show that C₂A mediated modulation of release probability can be bypassed through direct modulation of the C₁ domain, establishing that C₂A domain modulation is upstream of C₁ domain modulation. These data provide mechanistic insight in the action of Munc13 and specifically tests whether the complex set of modulatory functions of Munc13 are related and define their up or downstream relationship. We fully agree with the assessment of the reviewer that PDBu increases responses by increasing release probability. This can be nicely detected in in-cell experiments such as shown in Fig. 6d,e, which

allow acute changes in responses as small as 5%. The fact that the increase in response size does not lead to a significant increase in the p_{vr} of the Munc13-1 K32E mutant (Fig 6g), which has already an increased release probability, is simply due to the intrinsic variability that underlie the p_{vr} measurement in a cell-to-cell experiment, that makes difficult to detect changes in p_{vr} smaller than 20-30%. While the data did not reach significance, does not mean that the results are preliminary and they are fully consistent with the tested hypothesis, as all mutants reach similar release probabilities in the presence of phorbol esters, demonstrating that C_1 domain regulation is downstream of C_2A domain regulation.

5. The statistical analyses for the electrophysiological measurements are incorrect. Rather than T-tests, the authors should do one-way ANOVA (as they do for the EM) followed by post tests which correct for multiple comparisons, and they should make all relevant comparisons.

We performed the Kruskal-Wallis one-way analysis of variance followed by a multiple comparison Dunn's post hoc test, as rightfully suggested by the reviewer. Initially t-test was performed in figure 2 and 5 because each group of C_2A mutant data had its independent wildtype rescue and was considered as independent from each other. However, the ANOVA analysis will provide insights in the overall variability of the experimental settings. The revised statistics are now added to the figures 2 and 5 with the multiple comparison corrects. Only for figure 6 we keep the unpaired Student's T test: Mann-Whitney because the data from the treated group was independently compared with the non PDBu treated group

6. In the methods, it is stated that the resolution of the EM analysis is 2 nm. It is not explained where that number comes from, but the number is similar to the resolution in a paper by Imig et al (Neuron 2014), which used high pressure freezing combined with electron tomography. Given that the authors did not use tomography in the current study, it seems unlikely that they get this same (excellent) resolution, but the resolution is likely lower. This statement should be corrected.

The reviewer is correct that the resolution in the images is lower than in tomograms. We meant to state an approximation of the lower limit of our image analysis, which is manually performed using a modified version of the analysis software ImageJ. We have set the error for the manual analysis to approximately 3 times the pixel size (0.59 nm). We mainly define docking by an apparent direct contact of vesicle and plasma membrane when the two membranes are in direct contact without any apparent visual gap in between. We do not have a true estimation of actual resolution and therefore we have deleted this sentence from the methods section and rephrased the definition to better understand.

7. In respect to point 5: this is also important because it appears that Munc13 has two different docking roles: one mediated by the C2A domain, and an additional role that, in my view, is either (1) mediated by the MUN domain through SNARE complex assembly (a lot of literature from the Rizo, Rosenmund, Jorgensen labs) or by the other C2 domains (Liu/Rosenmund/Rizo et al, eLife 2016). It is possible that such differential docking roles could be revealed with resolution of 2 nm: for example a tethering function of the C2A domain may produce less tight docking, vs SNARE complex assembly may produce tighter docking. This topic deserves a more in-depth discussion.

The understanding of docking and tethering is rapidly evolving in the field and more sophisticated differences of vesicle distance vs. molecular configurations are likely to emerge. However, we do not claim to have the resolution to resolve those differences. As mentioned in our reply to point 6, we make a rather yes/no decision of vesicle docking based on the appearance of a gapless contact between vesicle and membrane. We therefore refrain from speculating based on our data set.

8. On the first page of the introduction, it is stated that "Lack of these proteins leads to a complete loss of docked and primed synaptic vesicles, as shown in Munc13-1/2 double deficient (DKO) mammalian neurons^{1,3-6}", The introduction of vesicle docking/tethering in the Munc13 KO deserves a more detailed discussion, as the phenotype strongly depends on the fixation method. In glutaraldehyde fixed samples, no docking deficit is observed (Augustin 1999, Varoqueaux 2002; in contrast to RIM mutants - Han 2011, Kaeser 2011). This is important because the docking role of the C2A domain may operate through RIM. But at the same time the data argue that it is not identical to the RIM phenotype, because if it were, it should be seen in glutaraldehyde. An additional statement at the end of the introduction also relates to this point ("This model is surprising given that RIM, like Munc13, is required for docking^{2,9-11}"). Previously used definitions of docking and effects of Munc13 and RIM deletions in the context of these definitions should be better introduced.

We have considerably revised the introduction with regard to the role of docking in Munc13 and now we emphasize the role of Munc13s in docking much more in the introduction page 3-4.

9. The color code in fig 6e is bad, it is not clear which one is WT and which one is K32E.

We appreciate this suggestion of the reviewer and we have changed now the color code mentioned in the figure 6 for the traces of the mutant K32E.

10. At the bottom of the first page of the introduction the roles for the MUN domain in priming are described. A recent paper (Liu/Rosenmund/Rizo eLife

2016) should be included, which is at odds with an isolated role for the MUN domain in priming.

We have included the additional reference which was not in press at the time of our paper submission.

11. It should be specified where in the RIM cDNA the RFP tag has been added.

We have included in the material and method section where the RFP was fused to RIM.

Reviewer #2:

1. Some of the interpretations are difficult to follow. Compared to Munc13 KO and WT rescue, docking is partly restored (~50%) by any form of Munc13 and yet their discussion states that docking is only intact in synapses expressing mutants that support heterodimer formation. They must be considering intact docking as being the 100% rather than the 50% value. Thus, one wonders why all of the mutant proteins (delta 1-520, EE, and KEE) support partial docking. Does this imply that another docking pathway involving Munc13 uses a mechanism independent of C2A and independent of RIM? The authors do discuss possible alternative pathways but this points out that the data set is difficult to cleanly interpret.

The reviewer is correct stating that we consider intact docking as a 100% of docked synaptic vesicles. To avoid misunderstandings with the 50% of docked synaptic vesicles that is observed in some of the mutants, we have deleted the expression “intact docking” and changed by “fully rescued docking”. At the same time, we have considered the difficulty to follow our interpretation on the effect of Munc13/RIM interaction in vesicle docking and we have rephrased the discussion (page 21-22) to introduce the possible alternatively models that can explain the remaining 50% docked vesicles observed. This point has been mentioned in more detailed in our response to the reviewer #1 in point 1.

2. Another data set difficult to interpret is that of Fig 5c where putative monomers of Munc13 seem to be as active in rescue as putative Munc13/RIM monomers. This is based on the interpretation of what these mutants represent but the triple mutant would seem to employ a RIM-independent mechanism that is nearly as strong as the RIM-dependent mechanism. There are probably many plausible interpretations involving other protein interactions.

We think the reviewer interpreted the data in Fig 5 that the Munc13-1 monomers are as active in rescue as the Munc13-1/RIM heterodimers. We apologize for misleading the reviewer. The data in figure 5 show that the Munc13/RIM heterodimers (K32E) are more active in rescuing vesicle priming (Fig. 5b),

evoked response (Fig. 5d) and in addition enhancing release probability (Fig 5e). These results are only partially compatible with Deng et al. 2011, which proposed that RIM is responsible for disrupting Munc13 homodimers and thus concluding that Munc13 monomers are fully active. We consider our findings as strong evidence that the Munc13-1/RIM complex is the most active in vesicle priming and release.

3. In rescue studies, it is important to know whether levels of expression of WT protein is sufficient to fully restore function during the expression period. In the current work, WT and mutant proteins are contrasted but whether WT protein fully restores function is not stated. This should be discussed. Does it matter how much of a protein is expressed during rescue or do the lentiviruses saturate protein expression?

In response to the rightful concerns of this and reviewer 1, we conducted a comparison of vesicle docking, priming, evoked release and vesicle release probability between Munc13-1/2 DKO neurons rescued with Munc13-1 WT and neurons from Munc13-1^(+/+)/Munc13-2^(-/-). Munc13-1 WT rescue construct largely restored the function of the endogenous protein by approximately 65-75% compared with Munc13-1^(+/+)/Munc13-2^(-/-) neurons (Supplementary Fig. 2b-c). We do think that the overall efficiency on the rescue of Munc13-1 WT is more than valid and to avoid possible errors we decide to normalized our data to the Munc13-1 rescue control group and not to the Munc13-1^(+/+)/Munc13-2^(-/-) neurons. See more detail in the answer for the reviewer #1 point 2.

Additionally, comparison of synaptic responses in Munc13-1^(+/+)/Munc13-2^(-/-), Munc13-1^(+/-)/Munc13-2^(-/-) and Munc13-1^(+/+)/Munc13-2^(-/-) plus Munc13-1 overexpression showed that neither reducing protein by 50%, in the case of the heterozygous neurons (Fig. 1c Varoqueaux et al. 2002), nor mild overexpression of Munc13-1 (Supplementary Fig. 2a) do not lead to significant changes in synaptic properties (Supplementary Fig. 2d). This suggests that synapses are not highly sensitive to overall Munc13-1 protein content variation, in particular given the apparent lack of a gain of function effect of Munc13-1. This part has been mentioned in more detailed in our response to the reviewer #1 in point 2.

4. In comparing WT and mutant protein expression, it is important to know whether the proteins are expressed at similar levels. In the current work, there is no indication of protein levels for WT or mutant proteins (in hippocampal cells). This may be of particular importance in this work because Munc13 levels are strongly depressed in RIM KO mice. Would failure to interact with RIM enable degradation of various mutant proteins? In Supplement, the authors detect somewhat variable expression of various mutants relative to WT protein in HEK cells. If this were also true in the hippocampal cells, would this affect the phenotypes?

5. In comparing WT and mutant protein expression, it is also important to know

whether the proteins localize normally or in a different manner. The authors present Munc13-GFP localization relative to VGLUT1 in Fig. 3 but the results are not discussed except to say that the Munc13-GFP proteins overlapped with the immunofluorescence signal for VGLUT1 suggesting proper localization. However, it appeared that K32E Munc13 (with or without RIM mutations) localized more broadly than the WT protein. The authors should quantitate co-localizations for these studies to determine whether similar localization is indeed observed, and the results should be discussed.

Response to the point 4 and 5:

In response to these two points, we performed additional immunocytochemistry experiments to study the expression levels and the localization of all truncation, deletion and point mutants Munc13-1. We generated two new Supplementary figures that are implemented in the paper. This part has been mentioned in more detailed in our response to the reviewer #1 in point 3.

Reviewers' comments:

Reviewer #1 (Remarks to the Author):

The authors have made reasonable effort to address the concerns that were raised and I support publication. That said, I suggest considering the wording in the manuscript on two points that relate to the previous round of review:

1. Expression levels and localization of the mutants, sup. Fig. 1: it is concluded on line 156 that expression levels were similar to wild type Munc13. Although I agree that it is unlikely that expression levels explain all effects, this experiment is only based on 5 cells (roughly 8 times less than for the electrophysiological experiments), but reveals strong, ~30% trends (S1B). Also, the sample images provided suggest that there may be strong differences in respect to synaptic vs. dendritic localization, with the C2A deletion mutants being much more dendritic. I strongly suggest adding a word of caution to the interpretation of those data.
2. Fig. 5g/lines 329-332: it is concluded that the E128.137K mutants facilitate, but the K32E mutant depresses more strongly. Here, however, despite recording 40-50 cells per condition, none of the effects are significant, and the magnitude of the trend in the E32K mutant is ~10%. The correct interpretation would be that there are no detectable changes in the PPR, being at odds with the Pvr calculations in 5e, which are indirect and rely on the ability to precisely quantify the sucrose response. The wording should better match the data.

Reviewer #2 (Remarks to the Author):

The authors provide extensive characterizations of Munc13-1 mutants in rescue studies with hippocampal neurons from Munc13-1/2 KO mice. The first set of studies focus on N-terminal deletion constructs that delete C2A and/or a proximal C-terminal segment prior to C1 from which the authors confirm a potential importance of C2A. In a second set of studies, C2A mutations K32E and E128K/E137K are used singly or in combination to dissociate dimeric Munc13, to prevent RIM interactions enabling dimeric Munc13 to dominate, or in combination to abolish self as well as RIM interactions to generate Munc13 monomer. The biophysical studies and HEK expression studies characterizing these mutants is well done and convincing. All mutants were tested in rescue studies for docking, RRP size, epsc responses to single APs, and release probabilities. Overall the quality of the experimental work is very high. However, there are few unambiguous or definitive conclusions that emerge from this complex dataset. Probably the key novelty is in the generation and study of RIM-binding mutants.

1. In the revised ms, the authors now provide critical control experiments needed for the rescue studies that were missing from the previous version of the manuscript. Key controls (synaptic localization of mutants) now shown strengthen the rigor of the data set but also pose new complications for the interpretation of a complex data set.

2. An important set of controls (Fig S2) showed the extent of rescue by WT and concluded that the precise level of WT Munc13 did not strongly affect rescue. That is fine for WT but may be different for a weaker Munc13 mutant. However, it is likely that synaptic localization is the most important parameter independent of protein levels per se.

3. It is quite difficult to precisely interpret the effects of 1-150, 151-520, and 1-520 deletion. The new data in Fig S1 suggest a trend toward a decrease in synaptic localization for each of these with means at 50% (although WT at 75% for comparison). This might be expected if C2A-RIM interactions help to localize Munc13 in synapses. If lowered values for the mutants were significant, it would suggest that

del 1-150 is not loss of function for docking and that del 151-520 is a strong gain of function for docking if normalized to synaptic localization. Synaptic localization would seem to be a key parameter that would need to be determined with greater precision to interpret effects on docking, priming etc.

4. The synaptic localization of the other mutants seemed to have been determined with greater precision. The synaptic localization of K32E is strongly increased relative to E128K/E137K (Fig S4). The authors conclude that "heterodimerization of the C2A domain with RIM optimizes Munc13's function in synaptic vesicle docking, priming and regulation of vesicular release probability", but it could instead reflect increased synaptic localization consistent with a proposed role for C2A-RIM interactions.

Reviewer #1:

“1. Expression levels and localization of the mutants, sup. Fig. 1: it is concluded on line 156 that expression levels were similar to wild type Munc13. Although I agree that it is unlikely that expression levels explain all effects, this experiment is only based on 5 cells (roughly 8 times less than for the electrophysiological experiments), but reveals strong, ~30% trends (S1B). Also, the sample images provided suggest that there may be strong differences in respect to synaptic vs. dendritic localization, with the C2A deletion mutants being much more dendritic. I strongly suggest adding a word of caution to the interpretation of those data.”

We fully agree that potential misexpression or mislocalization may contribute to the Munc13-1 mutants observed phenotypes. We performed additional immunocytochemistry experiments to determine with greater precision expression levels and synaptic localization of the truncated Munc13-1. These data now are the results of 3 different cultures with 10 cells per group per culture and 50 synapses per cell, resulting in approximately n=30 cells and 1500 synapses per group (Supplementary Fig. 1).

For the three different truncated Munc13-1 mutants we observed a clear correlation between levels at synapses of the Munc13 and the length of the protein expressed at equal volume of added virus, thus mutant constructs always had equal or higher expression than the wildtype.

(Supplementary Fig. 1b). The virus titer was identical to what was used for the electrophysiology recording in Figure 2. Therefore, the difference of rescue activity cannot be explained by a reduced expression of the protein in the synaptic compartment. We also performed genetic experiments using heterozygous Munc13-1 mutants and Wildtype neurons overexpressing Munc13-1 to assess the sensitivity of release to conditions when expression levels vary app. 50%, which did not revealed significant changes in Ca^{2+} evoked release or priming activity (Supplementary Fig. 2d).

We also analyzed the colocalization of the GFP and the VGLUT1 signals by comparing the Pearson correlation index in the three truncated mutants. Supplementary Fig. 1c shows now that the truncated mutants indeed have slight differences with respect to the WT and we observed a more diffuse axonal or dendritic signal in the deletion mutants lacking the C2A domain. We therefore agree with the reviewer, that in part, the phenotype may be caused by poor recruitment of the deletion mutants to the active zone within the nerve terminal. This has now been added to the Results (page 8) and Discussion (page 20).

2. Fig. 5g/lines 329-332: it is concluded that the E128.137K mutants facilitate, but the K32E mutant depresses more strongly. Here, however, despite recording 40-50 cells per condition, none of the effects are significant, and the magnitude of the trend in the E32K mutant is ~10%. The correct interpretation would be that

there are no detectable changes in the PPR, being at odds with the Pvr calculations in 5e, which are indirect and rely on the ability to precisely quantify the sucrose response. The wording should better match the data.

We thank the reviewer for raising concerns about the validity of the changes in release probability for Munc13-1 mutants that favor Munc13-RIM heteromerization. We performed additional experiments to strengthen the statistical power of the Pvr measurements of the Munc13-1K32E mutant. The data now show that release probability is not significantly increased (Fig. 5). We perform a re-analysis of our initial experiments and found two data points in the K32E data set that could be considered as outlier, producing an overestimation of the calculation of the release probability by sucrose. While we still consider, based on the sum of observations in the figures 1,4, 5 and 6, that the RIM-Munc13-1 heteromer does enhance efficiency of release, the statistical power of our experimental approach is not robust enough and we therefore removed this claim from the paper.

Reviewer #2:

“ 1. In the revised ms, the authors now provide critical control experiments needed for the rescue studies that were missing from the previous version of the manuscript. Key controls (synaptic localization of mutants) now shown strengthen the rigor of the data set but also pose new complications for the interpretation of a complex data set.”

2. An important set of controls (Fig S2) showed the extent of rescue by WT and concluded that the precise level of WT Munc13 did not strongly affect rescue. That is fine for WT but may be different for a weaker Munc13 mutant. However, it is likely that synaptic localization is the most important parameter independent of protein levels per se.

3. It is quite difficult to precisely interpret the effects of 1-150, 151-520, and 1-520 deletion. The new data in Fig S1 suggest a trend toward a decrease in synaptic localization for each of these with means at 50% (although WT at 75% for comparison). This might be expected if C2A-RIM interactions help to localize Munc13 in synapses. If lowered values for the mutants were significant, it would suggest that del 1-150 is not loss of function for docking and that del 151-520 is a strong gain of function for docking if normalized to synaptic localization. Synaptic localization would seem to be a key parameter that would need to be determined with greater precision to interpret effects on docking, priming etc.

Response to the point 1,2 and 3

In response to these points, we performed additional immunocytochemistry experiments to study the expression levels and the localization of all truncation,

and point Munc13-1 mutants. The results are incorporated in the Supplementary figures 1 and 4. As already stated in our response to reviewer 1, point 1, we included impaired recruitment of RIM binding deficient Munc13 mutants to the active zone as one of the possible interpretation of the effects in vesicle docking, priming and neurotransmitter release.

4. The synaptic localization of the other mutants seemed to have been determined with greater precision. The synaptic localization of K32E is strongly increased relative to E128K/E137K (Fig S4). The authors conclude that “heterodimerization of the C2A domain with RIM optimizes Munc13’s function in synaptic vesicle docking, priming and regulation of vesicular release probability”, but it could instead reflect increased synaptic localization consistent with a proposed role for C2A-RIM interactions.

We fully agree with the interpretation of the reviewer, as for responses for reviewer 1, and we performed additional immunocytochemistry experiments to determine with greater precision the expression levels and the localization at the synapses of the Munc13-1 point mutant. These data now are the results of 3 different cultures with n=30 cells and 1500 synapses per group (Supplementary Fig. 1 and Supplementary Fig. 4). Quantification of the intensities of the GFP show now that Munc13-1 (K32E) and (K32E,E128K,E137K) mutants have higher expression levels than the WT, while mutant Munc13-1 (E128K,E137K) showed a app. 20% reduction (Supplementary Fig. 4b). Since no significant difference in synaptic properties in heterozygous *Munc13-1^{+/+}/Munc13-2^{-/-}* neurons were found (Supplementary Fig. 2d), it is unlikely that these differences in expression levels change the release properties. But the examination of the degree of colocalization with VGLUT1 by Pearson correlation index in the three mutants displayed that the synaptic localization of the homodimerization-disrupting mutants Munc13-1 (K32E) increased relative to the WT, while the heterodimerization-disrupting mutants Munc13-1 (E128K,E137K) and (K32E,E128K,E137K) decreased. We also noticed, that the two mutants that interfere with the RIM binding, tended to show more diffuse signals in axons and neurites (Suppl. Fig 4).

REVIEWERS' COMMENTS:

Reviewer #1 (Remarks to the Author):

The authors have addressed my comments and I recommend acceptance.

Minor errors:

Fig. 5b&d: n's are shifted out of bars and are not readable

Line 359: 2011, not 2009.

Reviewer #2 (Remarks to the Author):

The previous review focused on concerns about whether the synaptic levels and localization of Munc13 mutants had been determined with sufficient precision and whether differences could account for differences in activities. The authors provide more extensive quantitation and introduce some qualifying statements into the text that take these findings into account.